# Long-term Brown Carbon and Smoke Tracer Observations in Bogotá, Colombia: Association to Medium-Range Transport of Biomass Burning Plumes

Juan Manuel Rincón-Riveros[1], Maria Alejandra Rincón-Caro[1], Amy P. Sullivan[2], Juan Felipe Mendez-Espinosa[1], Luis Carlos Belalcazar[3], Miguel Quirama Aguilar[1], and Ricardo Morales Betancourt[1]

[1]Civil and Environmental Engineering Department, Universidad de los Andes, Bogotá, Colombia
[2]Department of Atmospheric Science, Colorado State University, Fort Collins, CO, USA
[3]Universidad Nacional de Colombia, Bogotá, Colombia

**Correspondence:** Ricardo Morales Betancourt (r.moralesb@uniandes.edu.co)

**Abstract.** Light-absorbing aerosols emitted during open biomass burning (BB) events such as wildfires and agricultural burns have a strong impact on the Earth's radiation budget through both direct and indirect effects. Additionally, BB aerosols and gas-phase emissions can substantially reduce air quality at local, regional, and global scales, negatively affecting human health. South America is one of largest contributors to BB emissions globally. After Amazonia, the BB emissions from the wildfires and agricultural burns in the grassland plains of Northern South America (NSA) are the most significant in the region. However, few studies have analyzed the potential impact of NSA BB emissions on regional air quality. Recent evidence suggests that seasonal variations in air quality in several major cities in NSA could be associated with open biomass burning emissions, but it is still uncertain to what those sources impact air quality in the region. In this work, we report on 3 years of continuous equivalent Black Carbon (eBC) and Brown Carbon (BrC) observations at a hill-top site located upwind of the city of Bogotá and we demonstrate its association with MODIS detected fires in a 3000 km x 2000 km domain. Off-line $PM_{2.5}$ filter samples collected during three field campaigns were analyzed to quantify water-soluble organic carbon (WSOC), organic and elemental carbon (OC/EC), and biomass burning tracers such as levoglucosan, galactosan, and potassium. MODIS Active Fire Data and HYSPLIT back-trajectories were used to identify potential biomass burning plumes transported to the city. We analyzed the relationship between BrC, WSOC, water-soluble potassium, and levoglucosan to identify signals of regional transport of BB aerosols. Our results confirm that regional biomass burning transport from wildfires occurs annually during the months of January and April. The seasonality of eBC followed closely that of $PM_{2.5}$ at the city air quality stations, however, the observed seasonality of BrC is distinctly different to that of eBC and strongly associated to regional fire counts. The strong correlation between BrC and regional fire counts was observed both at daily, weekly, and monthly time-scales. WSOC at the measurement site was observed to increase linearly with levoglucosan during high BB periods, and to remain constant at $\sim 2.5\ \mu gC\,\mathrm{m}^{-3}$ during the low BB activity seasons. Our findings show, for the first time in this region, that aged BB plumes can regularly reach densely populated areas in the Central Andes of Northern South America. A source footprint analysis involving BrC

observations, back-trajectories, and remotely sensed fire activity shows that the eastern savannas in NSA are the main BB source region for the domain analyzed.

## 25   1   Introduction

Open biomass burning is a significant source of atmospheric aerosol particles and gas-phase pollutants (e.g., Bond et al., 2004; Aurell and Gullett, 2013; Tsimpidi et al., 2016). The particles emitted during biomass burning (BB) have a complex chemical composition dominated by primary organic matter (POM), elemental carbon (EC), and inorganic material such as sulfates, nitrates, and potassium (e.g, Yamasoe et al., 2000; Akagi et al., 2011). These species can contribute to deteriorated air quality
levels in urban centers (e.g., Phuleria et al., 2005; Garcia-Hurtado et al., 2014; Kollanus et al., 2016). The impacts of BB plumes over air quality for sites located several thousand kilometers away from the BB sources has been demonstrated (e.g., Forster et al., 2001; Cottle et al., 2014). Many studies have documented the negative effects of BB emissions over human health (Youssouf et al., 2014; Haikerwal et al., 2015; Reid et al., 2016). Additionally, the carbonaceous components of BB particles, which are typically internally mixed, contribute significantly to absorption of visible and UV light (Kirchstetter
et al., 2004). Elemental carbon is known to have a visible light absorption coefficient larger than that of any other aerosol component, and to substantially impact Earth´s radiation budget and climate. Due to its optical properties, EC is sometimes measured through light-absorption techniques, and when measured this way is referred to as equivalent Black Carbon (eBC) (Petzold et al., 2013). BC is the second largest contributor to anthropogenic radiative forcing with open burning of forests and savannas being the largest source (Stohl et al., 2015; Bond et al., 2013). The organic material (OM) present in aerosol
particles, mainly those produced in BB, biofuel combustion, and from other sources, has been shown to absorb light in UV wavelengths more efficiently than BC. The absorption increases proportionally to the amount of OM present in the aerosol (Yan et al., 2017; Mkoma et al., 2013). The collection of UV light-absorbing organic compounds present in aerosol particles are often termed Brown Carbon (BrC) (e.g., Kirchstetter et al., 2004; Andreae and Gelencsér, 2006; Wang et al., 2018), which is also a contributor to radiative forcing.
Biomass burning emissions from South America contribute the most to the global BC inventory, with 16% of the global emissions, surpassing those of other critical areas such as Asia and Africa (e.g., Koch et al., 2007; van der Werf et al., 2010). In particular, the Brazilian Amazonia and Cerrado regions produce substantial BB emissions, whose impacts have been the subject of numerous studies (e.g., Crutzen and Andreae, 1991; de Oliveira Alves et al., 2015; Gonçalves et al., 2016). Emissions from Amazonia and Cerrado occur typically between May and September, which corresponds to the dry season in the region
(Marengo et al., 2011). Fires in the savannas and tropical forests thousands of kilometers north of the Brazilian Amazonia, an area known as Northern South America (NSA), can also have significant global and local impacts (van der Werf et al., 2010). However, due to the significance of the BB emission from Amazonia, emissions from NSA have often been overlooked

despite its potential impacts on air quality and climate (Thornhill et al., 2017). The equatorward location of NSA causes its annual precipitation and BB emissions patterns to differ strongly from those of Amazonia. Peak emissions in the former occur between January and April with minimum BB activity between June and October. Those BB activity patterns in NSA are mostly determined by the dynamics of wet and dry seasons, which are in turn controlled by the annual south-north migration of the Intertropical Convergence Zone (ITCZ) (Pulwarty et al., 1998; Poveda et al., 2006; Mendez-Espinosa et al., 2019). Inter-annual variability in the intensity and length of the dry season, controlled by El Niño Southern Oscillation (ENSO), modulates the intensity of the peak BB emissions in NSA (Poveda et al., 2006).

The BB plumes generated in these fires can negatively impact the air quality experienced by over 60 million people that live in Venezuela, Colombia, and Ecuador. Only recently some studies have focused on the air quality impacts of BB emissions in this region. Observational studies performed at *Pico Espejo* mountain (Venezuela), over 4000 m in altitude, detected the passage of BB plumes during the dry season (Hamburger et al., 2013). Because of its vertical elevation, the *Pico Espejo* site often sampled free-troposphere aerosols, showing the potential long-range transport of aged BB plumes (Schmeissner et al., 2011). More recently, $PM_{2.5}$ and ozone observations in the sparsely populated savannas in NSA showed extremely high concentrations even in small towns where measurements were performed (Hernandez et al., 2019). These high $PM_{2.5}$ and ozone levels were associated with distant fires in the Venezuelan savannas. The potential regional-scale air quality impacts of BB emissions in NSA was recently explored by Mendez-Espinosa et al. (2019). In their work, a systematic analysis of air mass back-trajectories and MODIS hotspots for a ten-year period was conducted, indicating a strong association between fire counts in NSA and $PM_{2.5}$ concentrations in cities located hundreds of kilometers from the BB sources. Mendez-Espinosa et al. (2019) showed that BB emissions from the NSA savannas could be transported westward impacting air quality in several large metropolitan areas. However, there were no direct measurements of BB available to confirm the presence of BB aerosols in the urban areas considered. Since the main BB source regions are located hundreds of kilometers from the most densely populated areas, these BB plumes are likely aged. Atmospheric aging of BB aerosols has been shown to increase the oxidative potential of the particles (e.g., Wong et al., 2019a), potentially increasing the particles toxicity in addition to contributing to aerosol mass.

Detection of BB aerosols using chemical tracers is necessary to confirm the contribution of fires to aerosol loading at a given location. Traditionally potassium (K), levoglucosan, BrC, water-soluble organic carbon (WSOC), and other species have been used as biomass burning particle tracers (e.g., Sullivan and Weber, 2006a, b; Laskin et al., 2015; Shen et al., 2017; Martinsson et al., 2017). Potassium, K, has been extensively used as a BB tracer but there are significant non-biomass burning related sources of K, and it does not always correlate well with BB smoke (Pachón et al., 2013). Levoglucosan and other anhydrosugars, which are formed through the pyrolysis of cellulose, are more specific BB tracers (Simoneit et al., 1999). A potential limiting factor in the use of levoglucosan as a BB tracer is its oxidation in the atmosphere, with a lifetime of a few days when exposed to OH radical (Hennigan et al., 2010), reducing its abundance in long-range transported BB plumes that have aged in the atmosphere. Furthermore, Aerosol Mass Spectrometer data has shown that mass fractions associated with levoglucosan correlate strongly with light-absorbing carbonaceous material (e.g., Cubison et al., 2011; Lack et al., 2013). BB is also a significant primary source of WSOC (Sullivan and Weber, 2006b), but WSOC can also be formed through gas-to-

particle conversion of gas-phase organics (e.g., Weber et al., 2007). WSOC has also been shown to be a strong absorber in the UV part of the spectrum, as indicated by measurements of absorption Angstrom exponent from filter extracts (Hecobian et al., 2010). Because of its optically active components, BB aerosols can be detected through multi-wavelength particle light absorption measurements (e.g., Jeong et al., 2004).

In this work, we determine for the first time, to our knowledge, the presence of BB plumes in a large metropolitan area in NSA by using long-term observations of BB tracers. We linked the smoke tracer observations with regional BB activity, showing the role of medium-range transport of BB plumes in urban air pollution in Northern South America. We approach this problem by carrying out measurements on a hill-top site in Bogotá, Colombia. Continuous brown carbon and black carbon observations during a three-year period were used to establish temporal patterns in the BB tracer signal. The potential origin of the BB aerosols at the site was explored by analyzing the time series of MODIS active fire data in the NSA domain together with a systematic back-trajectory analysis (Mendez-Espinosa et al., 2019). Specific smoke tracers such as levoglucosan and water-soluble potassium were quantified. Our results show that smoke tracers in Bogotá are strongly associated with regional BB activity. The wildfires and agricultural burns in NSA from January to April contributes particularly to OC and WSOC concentration in the city of Bogotá.

## 2   Methods

We measured BrC and eBC continuously during a three year period at a hill-top site within the city limits of Bogotá, Colombia (Section 2.1). The site is known as the *Monserrate site*. Filter-based aerosol samples were also collected at this site over three different field campaigns spanning both, high and low BB activity in NSA. These samples were analyzed for smoke markers such as levoglucosan and other sugars. Water-soluble organic carbon (WSOC), inorganic ions, and EC/OC were also measured from these filter samples. Observations at our site were contrasted against those routinely collected at the Air Quality Monitoring Network of Bogotá (Section 2.5). Additionally, we combined MODIS active fire data with back-trajectory analysis to explore the potential transport of BB affected air masses by performing statistical association analysis between fire counts and smoke tracers concentrations (Section 2.4).

### 2.1   Measurement site description

The broader study domain covers a vast area of nearly 3.9 million $km^2$ (Figure 1a). The western part of NSA, dominated by the Andes mountain range, is a densely populated region with more than 60 million inhabitants. The eastern part of NSA includes the tropical grasslands and woodlands plains of the Orinoco river basin. The Orinoco river basin is sparsely populated and its economic activity is based on agricultural activities. The annual cycle of precipitation over the region is controlled by the meridional displacement of the ITCZ (Poveda et al., 2006). The ITZC southernmost location occurs typically during DJF. These months, are therefore characterized by dryer weather in NSA as the deep convection areas are displaced southward, towards Amazonia (Mendez-Espinosa et al., 2019). This mechanism in turn, largely explains the seasonality of BB activity in the region.

The measurement instruments were deployed at the *Monserrate* Sanctuary (Long. = -74.05649°, Lat. = 4.60582°). This Sanctuary is located on a hill-top on the eastern margin of the urban perimeter of Bogotá, Colombia (Figure 1b). The altitude of the *Monserrate* site is 3152 m above sea level, and 550 m above the mean height of the Andean plateau were the city of Bogotá lies (Figure 1). Easterly winds prevail at the site, placing it upwind from the densely populated metropolitan area with 9 million people (Figure 1b). According to the Air Quality monitoring stations in the city, annual average $PM_{2.5}$ concentration was 19 $\mu g\,m^{-3}$ during 2018, with a strong seasonal cycle in which monthly-mean $PM_{2.5}$ between February and March can reach 35 $\mu g\,m^{-3}$ and decrease to 11 $\mu g\,m^{-3}$ in July. Primary aerosol emissions are estimated to be 2600 tons/year, with a large contribution from diesel powered public transport buses and cargo trucks (Pachón et al., 2018). Road dust re-suspension emissions are highly uncertain, but are thought to contribute significantly to primary emissions (Pachón et al., 2018). There are no significant emission sources or urbanized areas east of the city (Figure 1b). Therefore, the Monserrate site location was intended to minimize the impact from the urban background and allowing the detection of regional signals. Wind speed and direction, UV radiation, relative humidity, and barometric pressure were also recorded on-site with a frequency of 10 minutes using a meteorological station Vantage-Pro2 (Davis Instruments, CA, USA.).

## 2.2 BrC and BC measurements

Aerosol light absorption coefficient, $b_{abs}$ ($Mm^{-1}$), was measured continuously using a 7-wavelength (370, 470, 520, 590, 660, 880, 950 nm), Aethalometer (Aerosol inc., model AE-33) described by Drinovec et al. (2015). The measurements were carried out at the *Monserrate* site during the three-year period from May 2016 to April 2019. Data was logged every 60 seconds. The sampling rate was 2 LPM through a $PM_{1.0}$ inlet (BGI model SCC0.732) to avoid potential mineral dust absorption interference on the measurements since most of either fresh or aged BB aerosol particles are in the sub-micrometer size range (Janhäll et al., 2010). The $b_{abs}$ raw data was corrected to account for filter loading effects (Virkkula et al., 2007). The loading correction parameter for each wavelength, $k_{\lambda}$, is automatically computed by asymmetrically splitting the sample flow and simultaneously measuring attenuation at two deferentially loaded filter spots (Drinovec et al., 2015). Absorption is also corrected for scattering using a multiple scattering parameter $C = 1.57$, i.e., $b_{abs} \rightarrow b_{abs}/C$. Equivalent black carbon concentration, eBC ($\mu g\,m^{-3}$), was computed from corrected $b_{abs}$ measured at the 880 nm channel. This wavelength is customarily used in Aethalometer measurements to define equivalent black carbon. At 880 nm the absorption from organics is minimized. Following the recommendations of Petzold et al. (2013), we report the mass absorption cross-section used to convert $b_{abs}$ to eBC. We used a mass absorption cross-section $\sigma = 7.77$ $m^2 g^{-1}$ , i.e., eBC$= b_{abs}/\sigma$. The estimated eBC limit of detection was 40 $ng\,m^{-3}$ for a 1 hour interval. A lower limit of detection is achieved with longer integration periods.

The spectral dependence of $b_{abs}$ was characterized with the Angstrom Absorption Exponent, $\alpha$, which is the logarithmic slope of the relation between $b_{abs}$ and wavelength, $\lambda$, i.e.,

$$\frac{b_{abs}(\lambda_1)}{b_{abs}(\lambda_2)} = \left(\frac{\lambda_1}{\lambda_2}\right)^{-\alpha} \tag{1}$$

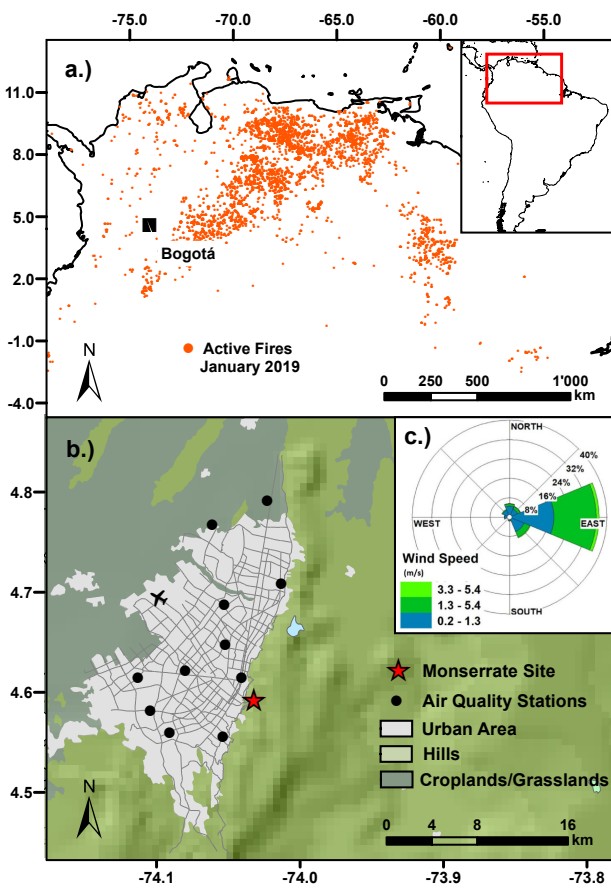

**Figure 1.** (a.) Geographic location of the Northern South America (NSA) domain as defined in this study and locations of MODIS hot-spots during January 2019. The inset shows the location of NSA relative to South America (b.) Location of the *Monserrate* site (star) near Bogotá and the AQ stations (filled circles). Colors in the map indicate land-use type, and the shading is showing terrain height variations. (c.) Wind direction observed at *Monserrate* site during the period May/2016 - Apr/2019

where $b_{abs}(\lambda_i)$ is the absorption coefficient at wavelength $\lambda_i$. Several different methods to apportion absorption to either fossil fuels or biomass burning have been developed (e.g., Sandradewi et al., 2008; Massabò et al., 2015; Chen et al., 2018). In this work, deconvolution of $b_{abs}$ between the contribution from fossil fuel and from biomass burning was done by applying the two-component model described by Sandradewi et al. (2008). In their model, aerosol absorption at any given wavelength is

155   separated into the contributions from BB and fossil fuel aerosols, i.e., $b_{abs}(\lambda) = b_{abs,BB}(\lambda) + b_{abs,FF}(\lambda)$. Furthermore, it is assumed that the spectral dependence of absorption for each component is characterized by a specific Angstrom exponent. This is, $b_{abs,BB} \sim \lambda^{\alpha_{BB}}$ and $b_{abs,FF} \sim \lambda^{\alpha_{FF}}$. Observational studies suggest that $\alpha_{FF} \approx 1$ (e.g., Sandradewi et al., 2008; Lack and

Langridge, 2013), however, there is a large variability in published Angstrom exponent values biomass burning aerosols (e.g., Hecobian et al., 2010; Harrison et al., 2013; Lack and Langridge, 2013; Kirchstetter et al., 2004).

The Angstrom exponent was computed using a wavelength in the near-UV, where absorption from organic compounds can be significant, and a near IR wavelength, where absorption is dominated by black carbon. However, as the 370 nm channel had a larger noise to signal ratio, the limit of detection of this channel was considerably higher, and was not used in the analysis. Equation 1 was then applied to $b_{abs}$ measured at 470 nm and 880 nm wavelengths to compute an observed $\alpha$. Sensitivity analysis were also carried out for $\alpha$ calculated between the 470 nm and the 950 nm channel. The fraction of light-absorbing aerosol attributable to BB, i.e., $f_{BB}$, is inferred from $\alpha$ by applying the two-component model (Sandradewi et al., 2008), i.e.,

$$f_{BB} = \frac{b_{abs,BB}(\lambda_1)}{b_{abs}(\lambda_1)} = \frac{\left(\frac{\lambda_1}{\lambda_2}\right)^{\alpha-\alpha_{FF}} - 1}{\left(\frac{\lambda_1}{\lambda_2}\right)^{\alpha_{FF}-\alpha_{BB}} - 1} \tag{2}$$

A detailed derivation of Equation (2) can be found in the Supplementary Material. We assumed $f_{BB}$ to be zero for $\alpha \leq \alpha_{FF}$ and one for $\alpha \geq \alpha_{BB}$. Another method to apportion absorption to sources which uses 5 wavelengths was tested (Massabò et al., 2015). However, this method was found to be more sensitive to assumed parameters than the simpler Sandradewi et al. (2008) used here.

Since BrC absorption results from the contribution of many different compounds, quantifying BrC concentrations from absorption measurements is challenging, as there is no single mass absorption cross section that can be applied. BrC mass concentration was estimated here as the fraction of absorption that is attributable to BB, computed as $BrC = eBC \times f_{BB}$. This is likely an underestimation of BrC as their mass absorption cross section is lower than that of eBC. We used the parameters $\alpha_{FF} = 1$, as has been suggested in several studies, and assumed a central value of $\alpha_{BB} = 2$. However, as there is significant uncertainty in $\alpha_{BB}$, we performed parameter sensitivity analyses by varying $\alpha_{BB} = 2.0 \pm 0.4$ and $\alpha_{FF} = 1.0 \pm 0.1$ (Sandradewi et al., 2008; Lack and Langridge, 2013; Harrison et al., 2013). The BrC estimates from optical absorption measurements are also compared with analytical quantification of levoglucosan and other BB combustion tracers (Section 2.3) known to be strongly related to BrC (Lack et al., 2013).

## 2.3 Biomass Burning Tracers

Filter-based aerosol samples were collected during three different field campaigns (Table 1) at the *Monserrate* site described in Section 2.1. The campaigns were designed to span high and low BB activity periods in NSA. Two of these campaigns were carried out during the high BB activity season (Campaigns 1 and 3) from January to April of 2018 and 2019, respectively, and one campaign was carried out during NSA rainy season (Campaign 2) from July to September 2018. Samples were collected onto 37 mm quartz filters for 24 hour periods (starting at midnight) every other day using a low-volume sampler. The sampler has a PM$_{2.5}$ inertial impaction stage and a sampling flow rate of 10 LPM. A total of 88 samples were collected and analyzed

to quantify BB tracers on the aerosol samples. Blank samples at each sampling site were also analyzed. These handling blank filters were carried to the sampling site and placed in the sampling instrument with the vacuum pump turned off.

**Table 1.** PM$_{2.5}$ sampling campaigns carried out on *Monserrate* Site, where $N$ stands for the number of filter samples

| Campaign ID | Sampling Period | N |
|---|---|---|
| 1. High-BB | 2018/01/15 - 2019/04/15 | 31 |
| 2. Low-BB | 2018/07/15 - 2019/09/15 | 24 |
| 3. High-BB | 2019/01/15 - 2019/04/15 | 33 |

190    Prior to its deployment for sampling, the quartz filters were pre-baked at 550°C for 12 hours to reduce their organic background and later placed in a desiccator to prevent water vapor absorption. After sampling, the filters were stored in a freezer at -80°C in plastic Petri dishes. At the end of each field campaign samples were sent over-night in a refrigerated container for chemical analysis to the Collett Laboratory at Colorado State University.

Organic carbon (OC) and elemental carbon (EC) were determined from a 1.4 $cm^2$ punch in each filter using a thermal/optical

195    transmission (TOT) EC/OC semi-continuous analyzer (Sunset Labs Inc.) following the NIOSH Method 5040 (Birch and Cary, 1996). The Limit of detection (LOD) was 0.2 $\mu$gC m$^{-3}$ and 0.5 $\mu$gC m$^{-3}$ for OC and EC, respectively. The remainder of the 37 mm filter was extracted in 15 ml of deionized water, and the extracts filtered with 0.2 $\mu m$ PTFE syringe filter to remove insoluble particles. Water-soluble organic carbon (WSOC) was measured with a TOC analyzer (Siervers Model M9 Turbo). This instrument measures WSOC by converting all organic carbonaceous material in the water extract to carbon dioxide using

200    chemical oxidation by ultraviolet light (UV) and ammonium persulfate. The LOD for WSOC in this study was 0.1 $\mu$gC m$^{-3}$. The overall measurement uncertainties for compounds analyzed is estimated to be $\sim 10\%$ (Sullivan et al., 2008).

A fraction of the aqueous extract was used to analyze for carbohydrates (including levoglucosan) through High-Performance Anion-Exchange Chromatography with Pulsed Amperometric Detection (HPAEC-PAD) (Sullivan et al., 2011). This technique uses a Dionex DX-500 series ion chromatograph with a Dionex GP-50 pump and a Dionex ED-50 electrochemical detector

205    operating in integrating amperometric mode using waveform A. Detailed descriptions of this method can be found elsewhere (e.g,. Sullivan et al., 2008, 2011). The LOD for carbohydrates quantification is less than $\sim 0.1 ng/m^3$.

Another portion of the aqueous extract was used to quantify inorganic anions and cations, including water-soluble potassium (WSK). For this analysis we used a Dionex ICS-3000 ion chromatograph with a conductivity detector, a isocratic pump and self-regenerating cation/anion suppressor. Cations were separated using a Dionex IonPac CS12A analytical column with a

210    flowrate at 0.5 mL $min^{-1}$ of 20 mM methanesulfonic acid eluent. The LOD for the various cations was 0.02 $\mu$g m$^{-3}$. In the case of anions, a Dionex IonPac AS11-HC anion-exchange column with a flowrate at 1.5 mL $min^{-1}$ of sodium hydroxide eluent was employed. The LOD for anions was 0.01 $\mu$g m$^{-3}$. This type of method has been applied by other studies (Tzompa-Sosa et al., 2016; Prenni et al., 2012) and further method details are presented by Sullivan et al. (2008).

## 2.4 Active Fires and Back-trajectory Analysis

MODerate-resolution Imaging Spectroradiometer (MODIS) observations were used to locate and count fires and its Fire Radiative Power (FRP) daily in the NSA domain (long.=-79.0°, lat.=-4.4°as bottom left; long.=-51.7°, lat.=13.1°as top right) during May 2016 to April 2019. Only those active fires labeled with a $\geq 75\%$ confidence level were included in the analysis (Justice et al., 2002). The spatial distribution of fires during January 2019 is shown in Figure 1a, where the substantial dry-season BB activity on the eastern savannas of the Orinoco river basin can be seen.

We constructed several time series of daily fire counts, $N_f$, in the NSA domain by applying a variety of criteria. In the simplest criterion, all active fire counts in the domain, $N_{f_{All}}$, with confidence $\geq 75\%$ were considered. Next, a set of distance criteria were applied and only the subset $N_{f,<R}$ of fire counts within a circular region with radius $R$ from Bogotá were included. Time series considering radii of 200 km, 400 km, 600 km, 1000 km, and 1500 km were built following this method. Similarly, additional time series of only those fires in an annular region, $N_{f,R_1-R_2}$, defined by distances $R_1$ and $R_2$ were considered, i.e., those fires within $R_1 < R < R_2$.

MODIS active fire data was combined with Lagrangian back-trajectory analysis to explore the potential transport of BB affected air masses following the methods of Mendez-Espinosa et al. (2019). Air-mass back trajectories arriving with three hour intervals (00:00, 03:00, 06:00, 09:00, 12:00, 15:00, 18:00 and 21:00 GMT-5) at the Monserrate site at 1000 m a.g.l., i.e., eighth daily, were computed using the NOAA HYSPLIT model (Stein et al., 2015; Draxler and Hess, 1998; Donnelly et al., 2015). Each trajectory was calculated for 96 hours prior to arrival in order to account for distant emission sources and avoid uncertainties on regional analysis due to longer trajectories (Donnelly et al., 2015). The Lagrangian trajectories were driven by GDAS1 meteorological fields which have an horizontal resolution of 1°x 1°(Su et al., 2015). The trajectory data was systematically analyzed using the OpenAir package (Carslaw and Ropkins, 2012) and the SplitR package of the open source programming language R. With this method, we constructed a time series of up-wind fire counts, $N_{f_{UpW}}$, using an algorithm to select only upwind fires according to the trajectory analysis (Mendez-Espinosa et al., 2019). For this, a buffer zone of 150 km was defined around each of the 96 hourly locations defining one of the eight back-trajectories reaching the receptor city in any given day. Then, only active fires in these buffer zones were included in the analysis. This method should account for any time lag between the occurrence of a fire and the effect over concentrations at a distant site. Additionally, time series of daily FRP data were constructed following the same procedures described here for $N_f$. A statistical association analysis was then carried out between between the data collected on-site, and the time series of fire counts and FRP. A source-footprint analysis was performed by combining the BrC observations, back-trajectories, and MODIS retrieved FRP (Supplementary Material).

## 2.5 PM$_{2.5}$ and eBC from the city monitoring stations

We retrieved PM$_{2.5}$ concentrations from the public air quality monitoring data repository of the Air Quality Monitoring Network of Bogotá. The air quality network data was used to contrast their magnitude and temporal patterns to those of the eBC and BrC observations at the Monserrate site. The PM$_{2.5}$ record from the air quality network covers the entire monitoring period. The air quality network has eleven stations across the city (Figure 1).

## 3    Results and Discussion

Multi-wavelength observations of $b_{abs}$ were used to apply Equation 2 as described in Section 2.2 to obtain eBC, absorption Angstrom exponent, $f_{BB}$, $b_{abs,BB}$ (at 470 and 880 nm), and the corresponding estimated BrC, all with hourly and daily temporal resolution. The complete time-series of daily-mean observations of eBC and BrC at the *Monserrate* site can be seen in Figure 2. Two short maintenance and calibration periods are seen in the data set as missing values. Mean total aerosol absorption $b_{abs}$ at 880nm for the observation period was $11.8 \pm 11.2\,\mathrm{Mm}^{-1}$. The large variability in the data occurs both at daily and monthly time scales. There is a marked seasonal cycle, with mean $b_{abs} = 15.0\,\mathrm{Mm}^{-1}$ for January to March (JFM) and $5.0\,\mathrm{Mm}^{-1}$ for June to August (JJA). Similarly, the mean absorption attributable to BB, $b_{abs,BB}(470nm)$, for the entire campaign was $2.41 \pm 2.87\,\mathrm{Mm}^{-1}$, with a statistically significant difference between the high BB burning activity periods (JFM) and those of low BB activity (JJA). The JFM $b_{abs,BB}(470nm)$ was $4.05\,\mathrm{Mm}^{-1}$ while the JJA mean was $0.71\,\mathrm{Mm}^{-1}$.

**Figure 2.** Time series of (a.) PM$_{2.5}$ ($\mu\mathrm{g\,m}^{-3}$) from Bogotá Air Quality Monitoring Stations, (b.) equivalent Black Carbon ($\mu\mathrm{g\,m}^{-3}$) from Monserrate Site (left axis is $b_{abs,880nm}$ in (Mm$^{-1}$)), (c.) Brown carbon ($\mu\mathrm{g\,m}^{-3}$) from *Monserrate* Site (left axis is $b_{bb,880nm}$ in (Mm$^{-1}$)), and (d.) Daily MODIS active fire counts in NSA within 600 km and 1000 km from Bogotá. Thick lines are seven-day moving averages from the original time series of daily-mean values. Faded lines in all panels show daily-mean values.

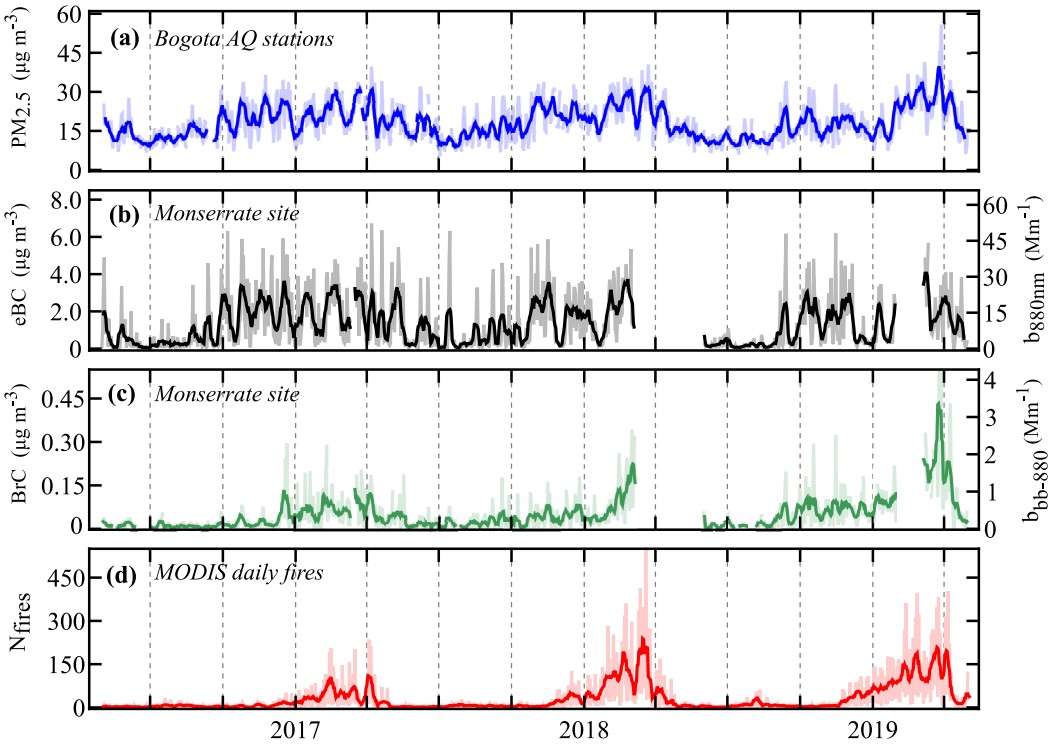

Accordingly, the observed daily-mean eBC concentration (Figure 2b) ranged from 0.02 to 5.0 $\mu\mathrm{g\,m}^{-3}$. Inferred BrC concentration were lower, with a maximum daily-mean of 0.44 $\mu\mathrm{g\,m}^{-3}$. The highest $f_{BB}$ was detected during the 2018-2019 dry

season (DJF), reaching up to a monthly mean of 15% for February. The campaign mean absorption Angstrom exponent was close to 1, indicating a strong influence from fossil fuel combustion sources. Observed $\alpha$ was found to vary according to the wavelength pair chosen in its calculation, i.e., $\alpha_{450nm-950nm} = 1.025 \pm 0.2$ and $\alpha_{450nm-880nm} = 1.065 \pm 0.22$. A sensitivity analysis on the $b_{abs,BB}(470nm)$ depending on the specific wavelengths chosen to calculate absorption Angstrom exponent is included in the Supplementary Material. Additionally, daily-mean $PM_{2.5}$ retrieved from the AQ monitoring stations is shown in Figure 2a, and $N_{f,600-1000}$ constructed according to Section 2.4 is shown in Figure 2d.

Day to day variations in $PM_{2.5}$ measured at the AQ monitoring network and eBC observed at the Monserrate site have a similar temporal pattern (Figure 2a and 2b). A simple linear correlation analysis between the two data sets using the Spearman correlation, $\rho_{PM_{2.5},BC}$, confirms this relationship as $\rho_{PM_{2.5},BC} = 0.76$. The high correlation between both datasets suggests that eBC at the Monserrate site is closely associated to urban emissions. According to a recent emission inventory in Bogotá, mobile and industrial emissions are the dominant primary combustion particle sources in the city. Furthermore, cargo and public transportation have the largest emissions share, and most of those vehicles are diesel powered (Pachón et al., 2018). To examine the degree of influence of the city emissions at the Monserrate site we analyzed mixed layer height from daily radiosondes data at the airport station (SKBO station). We found that the Monserrate site is typically above the mixed layer early in the morning, up until 9:30 am when the mixing layer expands surpassing the site altitude (Supplementary Material). Diurnal concentration patterns for eBC at the Monserrate Site and at the air quality monitoring stations support this hypothesis, since morning peak concentrations are observed with a lag of 1.5 to 2 hours at Monserrate site compared to the city AQ stations (Supplementary Material).

Contrastingly, the BrC observations in Figure 2 show a significantly different temporal structure compared to both $PM_{2.5}$ and eBC. A correlation analysis shows a substantially lower correlation, $\rho_{PM_{2.5},BrC} = 0.54$, compared to that of eBC and $PM_{2.5}$. This dissimilarity in the observed temporal patterns is indicative of a difference in the activity of sources of BrC and those of eBC, suggesting that the BrC signal is controlled by BB outside of the city.

## 3.1   Monthly-mean BrC and eBC

The annual cycle for eBC at the Monserrate site (Figure 3b) is similar to that of $PM_{2.5}$ registered at the air quality monitoring stations within the city (Figure 3a), with a bi-modal concentration pattern exhibiting maxima from February to March and from October to November. Part of the seasonal pattern in $PM_{2.5}$ has been previously explained by higher mixing heights during JJA and by lower mixing heights and increased static stability from December to March (Mendez-Espinosa et al., 2019). Monthly mean eBC at the Monserrate site ranges from 0.25 $\mu g\,m^{-3}$ in July to 1.70 $\mu g\,m^{-3}$ during February and November. Consistent with what is observed at a daily time-scale, the similarity between the annual eBC and $PM_{2.5}$ variations is expected as the site is within the urban mixed layer during most of the day and therefore, heavily impacted by urban traffic emissions. BrC seasonality at the Monserrate site, however, is distinctly different from that of either eBC or $PM_{2.5}$, with a lone maximum from February to April and not exhibiting a second peak in the last months of the year (Figure 3b). This discrepancy between the seasonal cycles of eBC and BrC strongly suggest that sources of both types of light-absorbing particles have different activity patterns along the year. Since eBC seems to be related to local emissions, the sources of BrC must be regional.

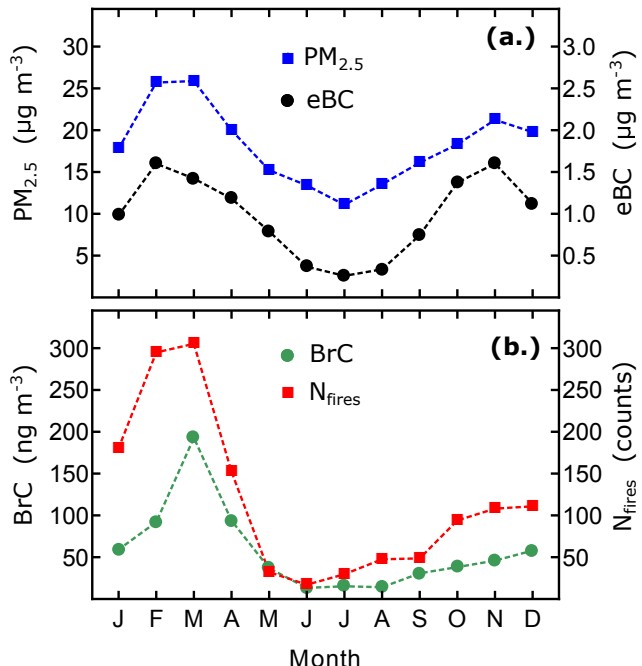

**Figure 3.** Monthly mean time series of Bogotá's PM$_{2.5}$ ($\mu$g m$^{-3}$), equivalent Black Carbon ($\mu$g m$^{-3}$), brown carbon ($\mu$g m$^{-3}$) and fire counts. BrC and eBC are measrued at the Monserrate site, while the PM$_{2.5}$ are observations from the air quality monitoring network. $N_f$ is the monthly mean of daily fire counts.

A potential explanation for the distinct seasonality of BrC at the site is found when analyzing the BB activity through MODIS active fire data after applying the fire counting algorithms described in Section 2.4. Figure 3b shows that the seasonality of BB activity is similar to that observed for BrC at the site, further suggesting a potential association between BrC measured in the Monserrate site and regional BB activity. Our observations are broadly consistent with other available studies of aerosol absorption in the region that have reported an increase in $b_{abs}$ and Angstrom exponent during the dry season. Observations at the ATTO tower in central Amazonia show $b_{abs,635nm} = 4.0 \pm 2.2$ Mm$^{-1}$ during the dry season (Saturno et al., 2018). Other observations at Pico Espejo, in NSA show $b_{abs,525nm} = 0.91 \pm 1.2$ Mm$^{-1}$ during dry season, corresponding to three times the mean value observed during the wet season. However, both sites correspond to locations near the source areas, while our observation site is an urban site far away from the main biomass burning areas.

### 3.2 Association with MODIS fire counts

To establish whether observed BrC at the Monserrate site is related to regional BB activity, we performed a systematic statistical association analysis between BrC observations and the different fire counting methods described in Section 2.4. The Spearman correlation for the daily-mean as well as the seven-day moving average for eBC and BrC with $N_f$ were calculated and summarized in Table 2. Overall, a weak statistical association was found between eBC and $N_f$, with values much lower

than those observed between eBC and PM$_{2.5}$, confirming that a large fraction of eBC measured at the site is likely from local fossil fuel combustion sources. Furthermore, regardless of the fire counting scheme applied, the statistical association between BrC and $N_f$ is stronger in all cases than that of eBC and $N_f$. Thus, despite the proximity of the measurement site to the city and the impact of local emissions, the association between regional BB activity in NSA and BrC suggest that the measurements at the site are able to differentiate the relatively small signal from regional BB to that of local emissions.

**Table 2.** Statistical association expressed through Spearman correlation between the different fire counting methods and eBC and BrC measured at Monserrate Site. Mov. Avg. is the Spearman correlation between smoothed time series with a seven-day moving average, and Daily are the Spearman correlations between daily-mean variables

| MODIS fire counts | BrC | | eBC | |
|---|---|---|---|---|
| | Mov. Avg. | Daily | Mov. Avg. | Daily |
| $600 < R < 1000$ km | 0.570 | 0.443 | 0.133 | 0.168 |
| $400 < R < 600$ km | 0.556 | 0.368 | 0.195 | 0.170 |
| $R < 1000$ km | 0.554 | 0.448 | 0.148 | 0.186 |
| All-fires (>75%) | 0.545 | 0.419 | 0.263 | 0.214 |
| $R < 600$ km | 0.521 | 0.369 | 0.167 | 0.163 |
| $200 < R < 400$ | 0.495 | 0.334 | 0.171 | 0.178 |
| $1000 < R < 1500$ | 0.454 | 0.251 | 0.095 | 0.035 |
| Up-Wind fires | 0.454 | 0.352 | -0.063 | -0.031 |
| $R < 400$ km | 0.453 | 0.316 | 0.114 | 0.152 |
| $R < 200$ km | 0.173 | 0.107 | -0.096 | 0.005 |

Table 2 is sorted according to the Spearman correlation between the seven-day moving average fire counts and BrC. The result of the analysis shows a stronger association between BrC and distant fires, and the weakest association for those fires within 200 km of Bogotá. This is likely due to the lower number of nearby fires compared to the abundant hot-spots in the savannas and tropical forests in NSA. Therefore, either at a daily or weekly time-scales, the concentration of UV absorbing carbonaceous material in Bogotá is more closely associated with regional BB activity than with local emissions. The fire counting method that only includes up-wind fires does not perform much better than the other methods considered. These results are consistent with an increase in the regional BB aerosol background during the dry season. A spatial footprint analysis of BB source areas shows that in the period from December to March, the savannas Orinoco river basin are the likely source regions impacting BrC at the Monserrate measurement site (see Supplementary Material). The travel time of the air masses from the savannas to the measurement site suggests that ageing of the organic aerosols can occur.

### 3.3 Brown Carbon and Smoke Tracers

The continuous BrC measurements described in Section 3.1 are a strong indicator of the enhanced presence of UV absorbing aerosols. However, due to the uncertainties in mass absorption cross sections, aerosol absorption measurements alone are not straightforward to translate into BB aerosol concentrations. To establish the relationship between the Aethalometer based BrC (Section 2.2) and analytical methods to quantify BB aerosols (Section 2.3), we compared 24-hour average BrC concentrations

with smoke markers levoglucosan, galactosan, WSK, and WSOC, as well as EC. The analysis was done for those specific dates where collocated filter-based samples and optical BrC observations were available, totalling 58 valid dates. A strong linear association was found between BrC and levoglucosan ($R^2 = 0.87$, slope $= 0.32$), with the linearity spanning the full range of measurements (Figure 4a), suggesting that the optical measurements of BrC are indeed a good BB tracer. However, recent observational studies of ambient BB particles have shown poor correlation between levoglucosan and brown carbon in aged BB plumes, likely due to BrC photobleaching and to the oxidation of levoglucosan (e.g, Wong et al., 2019b). The strong correlation in our samples might indicate atmospheric transport times of up to 2 days, consistent with the lifetime of levoglucosan (Hennigan et al., 2010) and that of BrC, recently estimated between 13 to 30 hours (Wong et al., 2019b). These atmospheric transport times are consistent with the strong association between BrC and MODIS active fires within 600 km of the sampling site (Table 2).

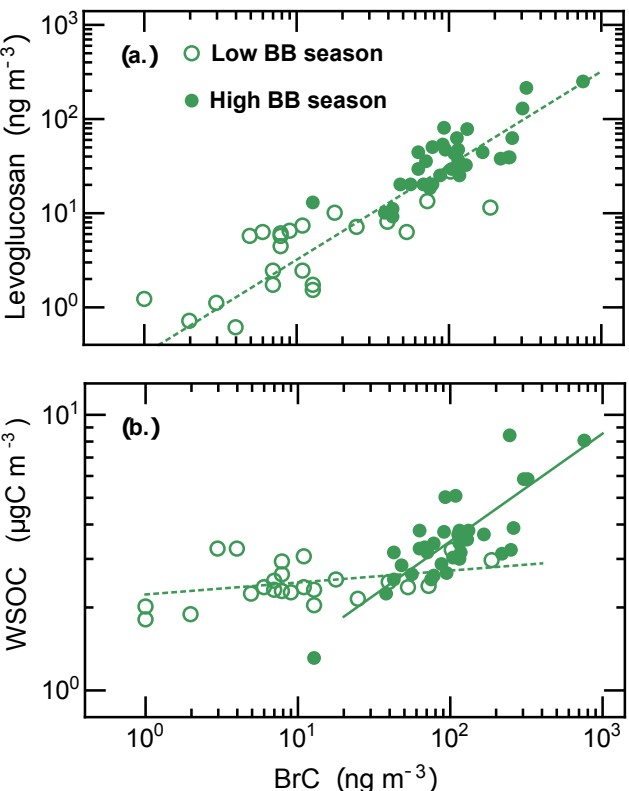

**Figure 4.** Scatter plot of daily-mean concentration of (a.) Levoglucosan and BrC, and (b.) WSOC and BrC, measured at the *Monserrate* site. Filled circles are samples collected during the high BB activity seasons, and open circles were collected during low BB activity.

Similarly strong associations were established for WSOC and other tracers (Table 3). When the association analysis is repeated only for data collected during the low-BB activity season (i.e., Campaign 2) the degree of association between BrC and WSOC drops significantly to just 0.34. This indicates that during low-BB activity months (JJA) WSOC has local sources

**Table 3.** Spearman correlation between optical measurements of carbonaceous aerosols brown carbon (BrC) and black carbon (eBC) and selected smoke tracers levoglucosasn (Lev.), galactosan (Gal.), WSOC, WSK, OC and EC measured at the Monserrate site. Correlation coefficients are shown for all data, as well as for the low and high BB seasons, respectively. $n$ represents the number of samples.

|  |  | Lev. | Gal. | WSOC | WSK | OC | EC |
|---|---|---|---|---|---|---|---|
| All-data (n=58) | BrC | 0.87 | 0.72 | 0.78 | 0.54 | 0.76 | 0.53 |
|  | eBC | 0.35 | 0.36 | 0.53 | 0.41 | 0.60 | 0.97 |
| High-BB (n=34) | BrC | 0.85 | 0.70 | 0.75 | 0.46 | 0.69 | 0.38 |
|  | eBC | 0.23 | 0.40 | 0.52 | 0.39 | 0.61 | 0.96 |
| Low-BB (n=24) | BrC | 0.66 | 0.60 | 0.34 | 0.13 | 0.52 | 0.88 |
|  | eBC | 0.51 | 0.45 | 0.31 | 0.13 | 0.35 | 0.78 |

likely not associated to BB. This is further supported by the much stronger association between levoglucosan and WSOC during high-BB seasons (0.73) compared to low-BB season (0.38). WSOC remained at values between 2 and 3 $\mu$gC m$^{-3}$, independent of BrC concentration, during the low-BB activity season (Figure 4b). However, it was seen to increase steeply as a function of BrC for the high-BB activity season. The mean WSOC observed for low BB activity was 2.5 $\mu$gC m$^{-3}$ while for high-BB activity period was 4.2 $\mu$gC m$^{-3}$ reaching a mximum daily-mean of to 8 $\mu$gC m$^{-3}$.

The observed levoglucosan concentrations are relatively low compared to what has been observed in other studies (e.g., Hecobian et al., 2010). However, the measured concentration of BB tracers are significant considering the distance between the measurement site and the source regions (see Supplementary Material).

## 4    Conclusions

In this study we determine for the first time the presence of medium-range transported biomass burning aerosols to urban areas in Northern South America by direct measurement of biomass burning tracers. The presence of BB burning affected air masses was confirmed by multi-wavelength optical measurements of brown carbon and black carbon, and by high-sensitivity detection of specific smoke tracers levoglucosan and galactosan. Continuous Brown Carbon measurements were performed during a three-year period with hourly time-resolution. These long-term observations allowed for the characterization of annual patterns in Black Carbon and Brown Carbon concentrations at the measurement site.

Despite the close proximity of the measurement site to the city center of a large, relatively polluted urban area, the statistical association between BrC and MODIS Active Fire data was strong on a daily basis. Furthermore, the association between BrC and fire counts was stronger for distant fires, i.e., those further than 400 km from the measurement site. This finding strongly supports the regional origin of the BB aerosol detected at the site. A source footprint analysis involving remotely sensed Fire Radiative Power data, back-trajectories calculations, and observed BrC concentration, further suggests that the eastern grasslands are the main biomass burning source region in NSA actually impacting populated urban areas. Our observations show that the annual pattern of Brown Carbon at the monitoring site was observed to have a single peak during February and March, coinciding with the peak in biomass burning activity in the region.

High-sensitivity levoglucosan, galactosan, and potassium measurements collocated with optical Brown Carbon observations were highly linearly correlated and showed excellent agreement. Therefore, the on-line optical observations at the measurement site were shown to be accurate tracers of BB aerosols when compared with well-established analytical methods. Water-soluble organic carbon (WSOC) was measured during high and low BB activity seasons. These observations suggest there are 2.5 $\mu$gC m$^{-3}$ of WSOC not related to BB , and that BB can contribute to WSOC, at which time can reach up to 8 $\mu$gC m$^{-3}$ for a 24 hour period.

The findings of this work demonstrate that background aerosol levels are increased every year due to the presence of biomass burning aerosols. The observed Brown Carbon and smoke tracer concentrations increase in close relation to the amount of MODIS detected fires. Despite the overwhelming black carbon signal coming from traffic emissions, a clear relation between the Brown Carbon signal and regional biomass burning aerosols is established. This results highlight that even distant biomass burning sources resulting from uncontrolled agricultural burns and deforestation negatively impact air quality in densely populated areas hundreds of kilometers away, and that they do so in a regular basis. During our observation period, the month with the largest contribution of BB aerosols to light-absorbing material was March with $10\% \pm 5\%$. The month with the largest load of BB aerosols was February of 2019, with $15\% \pm 6\%$. The uncertainty estimates in this fraction are due to uncertainty on the assumed absorption Angstrom exponent for biomass burning and fossil fuel burning used in the attribution algorithm.

*Data availability.* The data used in this article is available and will be provided upon request.

*Author contributions.* Conceptualization: R.M.B. and L.C.B., Investigation: J.M.R, M.A.R., M.Q.A. and A.P.S. Methdology: R.M.B., J.F.M. and A.P.S. Writing - Original Draft: J.M.R and R.M.B. Writing - review and editing: A.P.S., J.F.M. and L.C.B. Visualization: R.M.B., J.M.R. and M.A.R.

*Competing interests.* The authors declare that they have no conflict of interest.

*Acknowledgements.* This study was funded by the Colombian *Administrative Department of Science, Technology and Innovation - COL-CIENCIAS*, project No. 1204-745-56533 under grant contract No. FP44842-050-2017, and by the FAPA program from the Office of the vice-dean for Research from Universidad de los Andes. The authors thank the *Sanctuary of Monserrate* administration and Mons. Sergio Pulido Gutierrez for kindly allowing the display of measurement devices and for providing sustained logistic support throughout the measurement period.

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
