# Peer review of "Long-term Brown Carbon and Smoke Tracer Observations in Bogotá, Colombia: Association to Medium-Range Transport of Biomass Burning Plumes"

_Atmospheric Chemistry and Physics, 2019_

## Referee Comment (RC1) · Anonymous Referee #1 · 30 Jan 2020

Light-absorbing aerosols can affect both air quality and climate, so undersing their source and transport is important. This manuscript used a bunch of different observations to study the sources of light-absorbing aerosols over densely populated areas in the Central Andes of Northern South America. It showed that these aerosols are closely related to medium-range transport of biomass burning plumes. My comments are listed below.

Major comments

[Figure]

I am concerned about the uncertainty associated with the BrC and BC measurements reported in this work. As mentioned in the work and reported by many other studies, there is large variability in reported mass absorption cross-section and Angstrom exponent values for absorbing aerosols. However, this study still used a single certain value for these variables (i.e. =7.77 g/m3; FF=1; BB = 2), without estimating the uncertainty due to the variation of these values. I expect that both eBC and BrC concentrations would change a lot if one assumes different values for these optical parameters. In addition, the authors should also estimate the uncertainties resulting from the process of measuring and analyzing the biomass burning tracers.

Minor comments

Line 168: "The quartz filters were pre-baked at 550C for 12 hours to reduce their organic background and later placed in." why is it needed to be heated? Wouldn't it reduce the biomass burning semi-volatile OA?

Line 174. What is LOD?

Line 173-179. It seems OC and EC are measured in the same way? Then how does one differentiate OC from EC?

Line 236. "The similarity between both datasets shows that eBC measurements at the site are overwhelmingly dominated by EC emissions from urban traffic and industrial emissions". No absorbing OC emissions from urban traffic and industrial emissions?

Line 250. I think the major reason for the seasonal pattern in PM2.5 is the different emission source/strength in different seasons.

Line 263. I don't understand the reasoning here.

Line 302. Not clear to me how the authors get these numbers.

---

## Referee Comment (RC2) · Anonymous Referee #3 · 5 Feb 2020

This manuscript presents 3-year measurements of aerosol light absorption at multiple wavelengths over a site in the Northern South America (NSA) region. These measurements are combined with campaign-based biomass burning tracer measurements, MODIS fire counts and back-trajectory analysis to examine seasonal variations and source attributions of black carbon and brown carbon. It is one of the few observational studies over NSA, and clearly demonstrates the influences of nearby biomass burning on the local air quality in densely populated areas. The long-term observations of biomass burning aerosol properties are also useful in revealing the regional

and temporal variability in light absorbing aerosols. The sample collection and data postprocessing parts are well described.

My major concern is about the inference of brown carbon concentration in section 2.2. First, the assumptions of FF AAE (=1) and BB AAE (=2) are subject to large uncertainty. How sensitive are the derived BC and BrC concentrations to these assumed AAEs? It would be helpful to include some sensitivity analysis by varying the AAE values. Furthermore, lines 156-157 indicate that BrC concentration is computed as the product of eBC (equivalent BC concentration) and f_BB (fractional contribution of biomass burning to absorption). This is confusing: isn't the product equal to BC concentrations from the BB sources? How is it related to the BrC concentration? Presumably, BB aerosols should include both BC and BrC. But the inference method of BrC in section 2.2 seems to imply that absorption in BB aerosols is due to BrC. The calculation of BrC concentrations needs clarification.

Another suggestion is since there are previous studies of BrC from the Amazon BB region, it'd be interesting to compare the derived BrC loadings and absorption properties over NSA with those in discussions. That would help extend the findings in this study to a larger regional context.

Minor comments:

1. Line 38: the source of BrC is not limited to BB. They could also come from biofuel and biogenic sources. Suggest to revise the definition of BrC, i.e., Andreae and Gelencser, 2006

2. Lines 39-40: This sentence is inaccurate. The referred paper Bond et al., 2013 suggests that BC is the second largest contributors to anthropogenic radiative forcing, not BB particles.

3. Line 65: missing a comma after "...their work"

4. line 66: replace "finding" with "indicating"

5. Line 83: "Levoglucosan" doesn't need an initial capital letter

6. Line 92: brown carbon and black carbon do not need initial letter capitalized. This needs to be corrected in other places as well.

7. Line 123: W doesn't need capitalization

8. Line 126: what is Davis Advantage Pro II?

9. Figure 1 (b): suggest to add a color scale for the background map. Is it for terrain height?

10. Section 2.4: why not make the observatory site directly as the starting point of the back-trajectories, instead of Bogota? Since they are located at different altitudes.

11. Line 209: what is the spatial resolution of GDAS1 meteorology?

---

## Referee Comment (RC3) · Anonymous Referee #2 · 26 Feb 2020

This paper investigates the contribution of biomass burning from distant locations to air quality in Bogota Colombia based on an extensive data set of aerosol light absorption at multiple wavelengths. Most data reported are from a measurement site upwind and at higher elevation than the city. Filter measurements of smoke tracers are also used to support the analysis, along with satellite-based fire counts and air mass back trajectories. Overall the paper is a nice contribution to an understudied location and appropriate for publication in this journal. The results are interesting and the analysis very thorough, however, some components are confusing and should be clarified.

I agree with the other two reviewers that the sensitivity of the reported results to the choice of AAE for BC (AAE=1) and for BrC (AAE=2) should be assessed. A value of BrC AAE of 2 seems especially arbitrary. It is not clear to me why the authors utilized this analysis method at all since it adds unnecessary complexity and ambiguity; more related to this question follows below.

Why was 470 and 880 nm light absorption data used in the fractional biomass burning calculation? Explicitly state the reason. Eg, why not 370 and 950 nm, respectively? Similarly, why was 880 nm used for eBC, not 950 nm? Why not use all the wavelength data in some way, instead of just selecting a few wavelengths from the measurements (more on this below)?

Why does one even need to calculate a BC and BrC concentration, instead of just using the absorption coefficient? For example, simply using the absorption measured at a high wavelength as a tracer for BC and Abs measured at a low wavelength (e.g., 370, or if too noisy, 470 nm) as a tracer for BrC, after the Abs by BC at that wavelength is removed. This can be done by assuming a BC AAE of some value, such as 1. This seems like a much more transparent way to apportion BC and BrC from the multi-wavelength Aeth data and it eliminates the need to assume a characteristic BrC AAE. It also simplifies an uncertainty analysis on the sensitivity of the results to only the value of BC AAE. It would be interesting to see a correlation between the BrC mass inferred by the method in this paper and the BrC abs at some wavelength (eg, 370 nm).

Instead of picking a specific wavelength for BrC why not use all the Abs vs wavelength data. That is, fit the data with an AAE using all the wavelength and then use the fit to predict BrC AAE (data AAE-1) and then determine light absorption at some low wavelength with fit AAE-1.

Light absorption data are based on PM1, chemical composition and mass on PM2.5. PM1 was chosen to reduce possible influence of dust light absorption on the inferred BrC mass. The authors could test if there is any correlation between dust (eg, Ca2+)

[Figure]

and BrC.

Line 236-237: This line is unclear, suddenly there is a discussion that changes from eBC to EC. How does this data prove eBC is EC. Why not just say that eBC is from urban traffic and industrial emissions? Also, why is EC only assumed to be from these two sources?

Line 248-249, first line after heading 3.1. This line is unclear. Is the eBC, BrC and fire counts data (Fig 3b) from the hill top site and the PM2.5 mass (Fig 3a) from the urban air quality stations in the city? That means that Fig 3a has data from two different sites? This complicates the comparison and the discussion that follows this line. More clarity is needed here. Please specify on the plots in Fig 3 what site the data is from.

Line 282, typo change that to local emissions, to, than to local emissions.

Line 289-290 states, ... However, optical methods are not always quantitative methods to determine BB aerosol loading. What is this statement based on?

Line 314. Is this true; the Monserrate site (also called at times, the hill top site) maybe a fairly close distance to the urban center, but it is decoupled from the city at times due to its higher elevation and changes in BL height. This mixing of the hill top site with the urban site throughout the paper leads to confusion. Often the term monitoring site is also use, which is apparently the Monserrate site, not the urban air quality sites? I suggest being more specific and consistent throughout the paper on what the sites are called.

Last line of Conclusions. What is the 13% based on, mass ratio of eBC and BrC. This is then not an optical ratio and should be noted, it may also depend on how BrC was determined (AAE=2). Again, calculating mass concentrations of BC and BrC from the absorption data just leads to confusion and more uncertainty, in my view.

---

## Author Comment (AC1) · 9 Apr 2020

Response to Anonymous Referee #1

RC: Light-absorbing aerosols can affect both air quality and climate, so understanding their source and transport is important. This manuscript used a bunch of different observations to study the sources of light-absorbing aerosols over densely populated areas in the Central Andes of Northern South America. It showed that these aerosols are closely related to medium-range transport of biomass burning plumes. My comments are listed below.

Major comments RC: I am concerned about the uncertainty associated with the BrC and BC measurements reported in this work. As mentioned in the work and reported by many other studies, there is large variability in reported mass absorption cross-section and Angstrom exponent values for absorbing aerosols. However, this study still used a single certain value for these variables (i.e. =7.77 g/m3; FF=1; BB = 2), without estimating the uncertainty due to the variation of these values. I expect that both eBC and BrC concentrations would change a lot if one assumes different values for these optical parameters. In addition, the authors should also estimate the uncertainties resulting from the process of measuring and analyzing the biomass burning tracers.

Authors Response: We have now addressed the issue of uncertainty by performing sensitivity analysis on the parameters used in the calculations. Regarding the use of a (mass absorption cross section) MAC 7.77 g/m2 for eBC (at 880 nm), we would like to clarify that, by definition, it is necessary to assume a specific MAC to convert babs into a "Equivalent Black Carbon" concentration. We strictly followed the recommendations of Petzold et al., 2013 (Atmos. Chem. Phys., 13, 8365–8379, 2013) by explicitly stating the MAC used, so the calculation is transparent and reproducible. We now explicitly mentioned this in the manuscript. Furthermore, we have now included babs in Figure 2 by adding a secondary axis.

One significant issue was the lack of sensitivity to parameters. Figure R1 shows a sensitivity analysis performed on the parameters $\alpha$_FF and $\alpha$_BB. We computed the inferred BrC concentration for each set of parameters for high (DJF) and low (JJA) BB activity periods. Because our data is strongly influenced by urban emissions (dominated by traffic in Bogota) our observed Angstrom exponent is on average close to 1. Therefore, our deconvolution is much more sensitive to the assumed value of $\alpha$_FF than it is to the much more uncertain $\alpha$_BB. However, it should be noted that in all the parameter combinations the same trend remains, namely that during the high BB periods BrC is significantly higher than during JJA (i.e., has a strong seasonality). A

discussion in this regard is now included in the manuscript and the sensitivity analysis included in the Supplementary Material.

Minor comments RC: Line 168: "The quartz filters were pre-baked at 550°C for 12 hours to reduce their organic background and later placed in." why is it needed to be heated? Wouldn't it reduce the biomass burning semi-volatile OA?

Authors Response: The filters are pre-baked before being deployed for sampling. This is done exactly as the reviewer points out, to reduce semi-volatile OA from the filters, reducing this way any potential artifact during analysis post-sampling. We clarified this in the manuscript, and it reads "Previous to sampling, the quartz filters were pre-baked...."

RC: Line 174. What is LOD?

Authors Response: We intended LOD to stand for "Limit of Detection". We now explicitly define the term in the manuscript.

RC: Line 173-179. It seems OC and EC are measured in the same way? Then how does one differentiate OC from EC?

Authors Response: OC and EC are measured in the same instrument, with a technique called TOT (thermal-optical transmittance). However, they are not measured in the same way. The TOT measurement is based on the fact that the organic carbon contained in particles volatilizes at different temperatures. The organic carbon is defined in this technique as the carbon that becomes gas in a Helium atmosphere at temperatures below 580°C. Meanwhile, EC in this technique is defined as the fraction of carbon that does not volatilize after exposing it to 580°C, but that oxidizes when oxygen is added to the controlled atmosphere at temperatures above 580°C. The quantification of carbon in each case (either volatilized or oxidized) is done by converting it to CH4 to be detected with an FID.

We now expanded the explanation to avoid any potential confusion.

RC: Line 236. "The similarity between both datasets shows that eBC measurements at the site are overwhelmingly dominated by EC emissions from urban traffic and industrial emissions". No absorbing OC emissions from urban traffic and industrial emissions?

Authors Response: The phrasing was modified in this section. The phrase now reads "The strong correlation between both datasets suggests that eBC at the Monserrate site is closely associated to urban emissions. According to a recent emission inventory in Bogotá, mobile and industrial emissions are the dominant primary particle sources in the city. Furthermore, cargo and public transportation have the largest emissions share, and most of those vehicles are diesel powered (Pachón et al., 2018)."

Regarding the question of -No absorbing OC from urban traffic and industrial emissions? - It is possible (as has been recently show in the literature) that fossil fuels contribute to UV absorbing carbon (i.e., BrC). We acknowledge this in the paper now (in the introduction). However, EC is known to be the main absorber at near IR wavelengths, while OC from fossil fuel combustion is not a particularly strong absorber of near-IR light.

RC: Line 250. I think the major reason for the seasonal pattern in PM2.5 is the different emission source/strength in different seasons.

Authors Response: Indeed, as the reviewer points out, this is exactly our working hypothesis in this paper, namely that biomass burning emissions in the region increase PM2.5 concentration during the months of January-to-April, and we believe that we demonstrated that through measurements of biomass burning tracers in different seasons. In that specific paragraph we were merely pointing out that there are also meteorological conditions during those months (stronger surface inversions, stable conditions, lower mixing heights) that could concurrently have an impact of increasing PM2.5 concentrations (this is explained in the reference Mendez-Espinosa et al., 2019). Furthermore, there is no clear annual pattern in either public transport, cargo transport, or

industrial activities. There is no seasonal change in fuel composition as does occur in other countries.

RC: Line 263. I don't understand the reasoning here.

Authors Response: Point well taken. What we intended to say here was that the BrC we detected was likely aged biomass burning (because the sources are located hundreds of km away from our measurement site). The intended message is conveyed in the next section. Therefore, those lines were removed from the manuscript.

RC: Line 302. Not clear to me how the authors get these numbers. Authors Response: These numbers were obtained by averaging WSOC for high and low BB activity seasons respectively (i.e, those represented by the open and filled circles in Figure 4b). This now reads: "The mean WSOC observed for low BB activity was $2.5\mu gCm-3$ while for high-BB activity period was $4.2\mu gCm-3$ reaching up to $8\mu gCm-3$."

Please also note the supplement to this comment:
https://www.atmos-chem-phys-discuss.net/acp-2019-1124/acp-2019-1124-AC1-supplement.pdf

[Figure]

**Fig. 1.**

---

## Author Comment (AC2) · 9 Apr 2020

Response to Anonymous Referee #3

We kindly asked to reviewer to see attached file with the response to ensure all the symbols appear correctly.

RC: This manuscript presents 3-year measurements of aerosol light absorption at multiple wavelengths over a site in the Northern South America (NSA) region. These measurements are combined with campaign-based biomass burning tracer measure-

ments, MODIS fire counts and back-trajectory analysis to examine seasonal variations and source attributions of black carbon and brown carbon. It is one of the few observational studies over NSA, and clearly demonstrates the influences of nearby biomass burning on the local air quality in densely populated areas. The long-term observations of biomass burning aerosol properties are also useful in revealing the regional and temporal variability in light absorbing aerosols. The sample collection and data postprocessing parts are well described. My major concern is about the inference of brown carbon concentration in section 2.2. First, the assumptions of FF AAE (=1) and BB AAE (=2) are subject to large uncertainty. How sensitive are the derived BC and BrC concentrations to these assumed AAEs? It would be helpful to include some sensitivity analysis by varying the AAE values.

Authors Response: We have now included a sensitivity analysis showing how the uncertainties associated to these parameters impact the calculated attribution of absorption to combustion of biomass or fossil fuels (Figure R1). We also enhanced the discussion on the sources of uncertainty. In a now expanded supplementary material, we show the impact of parameter choice on the inferred BrC concentration. After performing this analysis, we showed the estimated BrC is only slightly affected by the choice of $\alpha\_BB$, while it is more sensitive to $\alpha\_FF$. However, in any case, the correlation of BrC and MODIS fire counts remains unchanged. Figure R1 and a subsequent discussion on the sensitivity is now included in the Supplementary Material.

For our specific data set, heavily influenced by traffic emissions , the observed angstrom exponent is closer to 1 ($\alpha\_(450nm-950nm)=1.025\pm0.2$ and $\alpha\_(450nm-880nm)=1.065\pm0.22$). Therefore, the inferred BB fraction is much more sensitive to $\alpha\_FF$ than it is to $\alpha\_BB$. This can be seen in the Figure R1 of this response (which has also been included in the Supplementary material for the final manuscript). This is positive for our study, as it is well known that $\alpha\_BB$ is much more uncertain than $\alpha\_FF$ (which is largely accepted to be close to 1). This analysis is now included in the manuscript.

RC: Furthermore, lines 156-157 indicate that BrC concentration is computed as the product of eBC (equivalent BC concentration) and f_BB (fractional contribution of biomass burning to absorption). This is confusing: isn't the product equal to BC concentrations from the BB sources? How is it related to the BrC concentration? Presumably, BB aerosols should include both BC and BrC. But the inference method of BrC in section 2.2 seems to imply that absorption in BB aerosols is due to BrC. The calculation of BrC concentrations needs clarification.

Authors Response: A section was included in the supplementary material to expand and clarify the decomposition method applied in our study. The Methods section was also expanded to improve clarity. The method we used (Sandradewi et. al. 2008) is often referred to as the "Aethalometer model". In our manuscript (section 2.2), absorption at any given wavelength is indeed considered to be due both to BB and FF at any given wavelength. The FF contribution is associated with a $\lambda^{(-1)}$ component (typical of BC rich FF sources) and the BB component is associated with an Angstrom exponent >1 (typical of sources with light absorbing OC, such as BB). Our approach, as suggested by the manufacturer, is to use optical properties of black carbon to estimate mass. This is likely an underestimation of true BrC mass as most studies suggest its mass absorption cross sections is lower than that of BC.

RC: Another suggestion is since there are previous studies of BrC from the Amazon BB region, it'd be interesting to compare the derived BrC loadings and absorption properties over NSA with those in discussions. That would help extend the findings in this study to a larger regional context.

Authors Response: Done. We included some new references were absorption measurements of BB aerosols are made in NSA. These include (Saturno et al., ACP, 2018; Hamburguer et al, ACP, 2013). The addition now reads "Our observations are broadly consistent with other available studies of aerosol absorption in the region that have reported an increase in babs and Angstrom exponent during the dry season. Observations at the ATTO tower in central Amazonia show babs,635nm= 4.0±2.2Mm−1 during

the dry season (Saturno et al., 2018). Other observations at Pico Espejo, in NSA show babs,525nm= 0.91±1.2Mm−1 during dry season, corresponding to three times the mean value observed during the wet season. However, both sites correspond to locations near the source areas, while our observation site is an urban site far away from the main biomass burning areas".

Minor comments:

Line 38: the source of BrC is not limited to BB. They could also come from biofuel and biogenic sources. Suggest to revise the definition of BrC, i.e., Andreae and Gelencser, 2006

Authors Response: Point well taken. We rephrased and reorganized this section to acknowledge other sources of BrC. It now reads: "The organic material (OM) present in aerosol particles, mainly those produced in BB, biofuel combustion, and from other sources, has been recently shown to absorb light in UV and short visible wavelengths more efficiently than BC. The absorption increases proportionally to the amount of OM present in the aerosol (Yan et al., 2017; Mkoma et al.,2013). The collection of UV light-absorbing organic compounds present in aerosol particles is often termed Brown Carbon (BrC) (e.g., Kirchstetter et al., 2004; Andreae and Gelencsér, 2006; Wang et al., 2018), which is also a contributor to radiative forcing."

RC: Lines 39-40: This sentence is inaccurate. The referred paper Bond et al., 2013 suggests that BC is the second largest contributors to anthropogenic radiative forcing, not BB particles

Authors Response: This oversight is now corrected in the manuscript. We now use the reference more accurately. It now reads: "Due to its optical properties, EC is sometimes measured through light-absorption techniques, and when measured this way is referred to as equivalent Black Carbon (eBC) (Petzold et al., 2013). BC is the second largest contributor to anthropogenic radiative forcing whit open burning of forests and savannas being the largest source (Stohl et al., 2015; Bond et al., 2013)."

RC: Line 65: missing a comma after "...their work"

Authors Response: Corrected.

RC: line 66: replace "finding" with "indicating"

Authors Response: Corrected.

RC: Line 83: "Levoglucosan" doesn't need an initial capital letter

Authors Response: Corrected.

RC:Line 92: brown carbon and black carbon do not need initial letter capitalized. This needs to be corrected in other places as well

Authors Response: This is now corrected throughout the manuscript.

RC: Section 2.4: why not make the observatory site directly as the starting point of the back-trajectories, instead of Bogota? Since they are located at different altitudes.

Authors Response: The (lat, lon) coordinates used in the calculation are indeed those of the Monserrate site. This typo is now corrected. However, it should be noted that the spatial resolution of the meteorological data (1 degree, roughly equivalent to 110 km) is too coarse to accurately represent differences in back-trajectories starting from nearby points. In a previous study, we found that due to the complex topography of the region, selecting starting points that are too close to or at the surface yields unrealistic back-trajectories. That is why we selected our arriving point at 1000 m.a.g.l, so it is not at the surface but remains within the mixing layer. This is now explicitly stated.

AC: Line 123: W doesn't need capitalization Authors Response: Corrected.

AC:. Line 126: what is Davis Advantage Pro II?

Authors Response: This is now corrected. The Vantage-Pro2 (it was erroneously typed in the original manuscript) is the specific model of the meteorological station used for the data collection, which is made by Davis Instruments. This is now explicitly written

in the manuscript.

AC:. Figure 1 (b): suggest to add a color scale for the background map. Is it for terrain height? Authors Response: Point well taken. The figure was modified and included a more descriptive legend (See Figure R2 included in this response). The color scheme is related to land-use cover (urban area, hills, and cropland/grassland). The shading is intended to qualitatively show terrain height variations, and this is mentioned in the caption. If height contour levels are included the plot gets cluttered and then is no longer effective.

RC: Line 209: what is the spatial resolution of GDAS1 meteorology? Authors Response: This is now corrected. GDAS1 meteorology is 1° x 1°. We now explicitly mention this in the manuscript.

Please also note the supplement to this comment:
https://www.atmos-chem-phys-discuss.net/acp-2019-1124/acp-2019-1124-AC2-supplement.pdf

[Figure]

[Figure]

**Fig. 1.** Figure - R2. Modified figure including a more detailed legend

---

## Author Comment (AC3) · 9 Apr 2020

RC: This paper investigates the contribution of biomass burning from distant locations to air quality in Bogota Colombia based on an extensive data set of aerosol light absorption at multiple wavelengths. Most data reported are from a measurement site upwind and at higher elevation than the city. Filter measurements of smoke tracers are also used to support the analysis, along with satellite-based fire counts and air mass

back trajectories. Overall the paper is a nice contribution to an understudied location and appropriate for publication in this journal. The results are interesting and the analysis very thorough, however, some components are confusing and should be clarified. I agree with the other two reviewers that the sensitivity of the reported results to the choice of AAE for BC (AAE=1) and for BrC (AAE=2) should be assessed. A value of BrC AAE of 2 seems especially arbitrary. It is not clear to me why the authors utilized this analysis method at all since it adds unnecessary complexity and ambiguity; more related to this question follows below.

Authors Response: This comment is common to all reviewers. We addressed the issue of uncertainty in two ways: 1- By expanding and explaining the uncertainty associated to the attribution of absorption to BB and FF (including sensitivity analysis) and 2- by discussing potential uncertainties associated to transforming babs into eBC and BrC. Figures were modified to include an axis with babs (in Mm-1) We also expanded the supplementary material to clarify and make more transparent how the separation between BB and FF was performed.

RC: Why was 470 and 880 nm light absorption data used in the fractional biomass burning calculation? Explicitly state the reason. e.g, why not 370 and 950 nm, respectively? Similarly, why was 880 nm used for eBC, not 950 nm? Why not use all the wavelength data in some way, instead of just selecting a few wavelengths from the measurements (more on this below)?

Authors Response: We did not use data from the 370 nm channel since its noise to signal ratio is higher than that of other channels. To analyze BrC/BC typically a short wavelength and a near-IR wavelength are used. The choice of 880 nm for BC was done as this is what has been historically reported in previous aethalometer studies. BC at 880 and 950 nm channels is almost identical for our data. This is now explicitly mentioned in the manuscript: "The Angstrom exponent was computed using a wavelength in the near-UV, where absorption from some organic compounds can be significant, and a near IR wavelength, where absorption is dominated by black carbon. However,

as the 370 nm channel had a larger noise to signal ratio, the limit of detection of this channel was considerably higher and was not used in the analysis. Equation 1 was then applied to babs measured at 470 nm and 880 nm wavelengths to compute an observed $\alpha$."

We also performed some sensitivity analysis on the pair of channels used to calculate the Angstrom exponent and evaluated its impact on our results. The mean Angstrom exponent varies slightly according to the wavelength pair chosen ($\alpha$_(450nm-950nm)=1.025±0.2 and $\alpha$_(450nm-880nm)=1.065±0.22) affecting the absolute value of inferred f_BB and BrC. However, because most of our analyses are based on correlations and associations to fires, these remain unchanged.

RC: Why does one even need to calculate a BC and BrC concentration, instead of just using the absorption coefficient? For example, simply using the absorption measured at a high wavelength as a tracer for BC and Abs measured at a low wavelength (e.g., 370, or if too noisy, 470 nm) as a tracer for BrC, after the Abs by BC at that wavelength is removed. This can be done by assuming a BC AAE of some value, such as 1. This seems like a much more transparent way to apportion BC and BrC from the multiwavelength Aeth data and it eliminates the need to assume a characteristic BrC AAE. It also simplifies an uncertainty analysis on the sensitivity of the results to only the value of BC AAE. It would be interesting to see a correlation between the BrC mass inferred by the method in this paper and the BrC abs at some wavelength (eg, 370 nm).

Authors Response: We are aware of the challenges of performing this decomposition. The method proposed by the reviewer is plausible, but it disregards contributions to absorption from BrC even at high wavelengths. The method we employed accounts for the contribution of BrC and BC absorption at all wavelengths. We now discuss this much more thoroughly in the manuscript. To address the valid concern regarding calculations of BC and BrC, we now added babs (Mm-1) in a secondary axis in Figure 2b and 2c. Since eBC and BrC are both proportional to babs, their correlation to the other datasets remains unchanged.

RC: Instead of picking a specific wavelength for BrC why not use all the Abs vs wavelength data. That is, fit the data with an AAE using all the wavelength and then use the fit to predict BrC AAE (data AAE-1) and then determine light absorption at some low wavelength with fit AAE-1.

Authors Response: During the data analysis phase, we did explore a variety of multi-wavelength methods. However, most of those methods require and even greater number of assumed parameters (e.g., Massabó et al., 2015). In their approach 5 wavelengths are used, but there is the need to assume three parameters and solve for three more. We tested this method and when applied to our data was much more sensitive to parameters choice than the simpler method we employed.

RC: Light absorption data are based on PM1, chemical composition and mass on PM2.5. PM1 was chosen to reduce possible influence of dust light absorption on the inferred BrC mass. The authors could test if there is any correlation between dust (eg, Ca2+) and BrC.

Authors Response: This is an excellent suggestion. However, we do not have Ca2+ data available in our samples. We think there is strong evidence showing that our BrC observations are strongly linearly correlated with levoglucosan and other BB tracers, suggesting its origin is indeed from biomass burning.

RC: Line 236-237: This line is unclear, suddenly there is a discussion that changes from eBC to EC. How does this data prove eBC is EC. Why not just say that eBC is from urban traffic and industrial emissions? Also, why is EC only assumed to be from these two sources?

Authors Response: Corrected. The phrase now reads "The strong correlation between both datasets suggests that eBC at the Monserrate site is closely associated to urban emissions. According to a recent emission inventory in Bogotá, mobile and industrial emissions are the dominant primary particle sources in the city. Furthermore, cargo and public transportation have the largest emissions share, and most of those vehicles

are diesel powered (Pachón et al., 2018)."

RC: Line 248-249, first line after heading 3.1. This line is unclear. Is the eBC, BrC and fire counts data (Fig 3b) from the hill top site and the PM2.5 mass (Fig 3a) from the urban air quality stations in the city? That means that Fig 3a has data from two different sites? This complicates the comparison and the discussion that follows this line. More clarity is needed here. Please specify on the plots in Fig 3 what site the data is from.

Authors Response: This is correct. We tried to be as clear as possible in the caption and throughout the text. Now Figure 2 in the manuscript has been modified to include the origin of the data (i.e., panels (b) and (c) are marked Monserrate site). Similar changes were performed on Figure 3. However, we want to emphasize to the reviewer that the aim of Figures 2 and 3 in the paper is to show that our BC measurements are strongly correlated to PM2.5 in the city, while BrC measurements do not. Instead, BrC resemble regional fire counts according to their correlation coefficients.

RC: Line 282, typo change that to local emissions, to, than to local emissions.

Authors Response: Corrected.

RC: Line 289-290 states, ... However, optical methods are not always quantitative methods to determine BB aerosol loading. What is this statement based on?

Authors Response: The statement originally meant to refer to the issue of translating absorption coefficient data into concentrations (which, as the reviewer pointed out, require the assumption of a MAC). That is what we originally meant by "quantitative". This whole paragraph is now rewritten and now reads: "However, due to the uncertainties in mass absorption cross sections, aerosol absorption measurements are not always straightforward to translate into BB aerosol concentrations. To establish the relationship between our Aethalometer based BrC (Section 2.2) and analytical methods to quantify BB aerosols (Section 2.3), we compared…."

RC: Line 314. Is this true; the Monserrate site (also called at times, the hill top site)

maybe a fairly close distance to the urban center, but it is decoupled from the city at times due to its higher elevation and changes in BL height. This mixing of the hill top site with the urban site throughout the paper leads to confusion. Often the term monitoring site is also use, which is apparently the Monserrate site, not the urban air quality sites? I suggest being more specific and consistent throughout the paper on what the sites are called.

Authors Response: Point is well taken. Our monitoring site is now referred to as Monserrate Site consistently throughout the whole manuscript. We also included a paragraph in the Methods to make explicit that our data comes from two different sources: our station at Monserrate (for eBC, BrC, and smoke tracers), and the AQ monitoring sites in the city (for PM2.5 only).

RC: Last line of Conclusions. What is the 13% based on, mass ratio of eBC and BrC. This is then not an optical ratio and should be noted, it may also depend on how BrC was determined (AAE=2). Again, calculating mass concentrations of BC and BrC from the absorption data just leads to confusion and more uncertainty, in my view.

Authors Response: The 13% is based on f_BB. With the expanded and improved Methods and Supplementary materials we show that f_BB=(b_(abs,BB) ($\lambda$))/(b_abs ($\lambda$)), i.e., the ratio of b_(abs,BB) ($\lambda$) We performed a sensitivity analysis of this monthly mean f_BB with $\alpha$_BB=2.0$\pm$0.4 and $\alpha$_FF=1.0$\pm$0.1. We know report a range in this percentage. It now reads: "During our observation period, the month with the largest contribution of BB aerosols to light-absorbing material was March with 10%$\pm$5%. The largest contribution was identified for February and March 2019, with 13%$\pm$6%. The uncertainty estimates in this fraction are due to uncertainty on the assumed absorption Angstrom exponent for biomass burning and fossil fuel burning used in the attribution algorithm"

Please also note the supplement to this comment:
https://www.atmos-chem-phys-discuss.net/acp-2019-1124/acp-2019-1124-AC3-

supplement.pdf

---

## Author Comment (AC4) · 9 Apr 2020

We thank the three anonymous reviewers for their thoughtful feedback, and constructive comments, which undoubtedly helped to improve our manuscript. The three reviewers accurately pointed out that the manuscript was missing an uncertainty analysis associated with the reported eBC and BrC concentrations, and they all emphasized the need to perform a sensitivity analysis for the parameters involved in the attribution of BrC and Black Carbon, to make the calculations more transparent. We addressed these issues (and all the other comments). This particular issue was addressed in the

following way:

1. Detailed description showing step-by-step the decomposition of absorption measurements (babs) due to fossil fuel (FF) and biomass burning (BB).

2. Detailed discussion of uncertainties associated with our approach, both, from the decomposition into FF and BB (assumed angstrom exponents), and from uncertain values of mass absorption cross sections

3. Sensitivity analysis to parameter choices.

All the additional comments were also addressed and incorporated in a new version of the manuscript. Detailed responses to each one of the reviewers' comments are detailed below (in blue) sorted by publishing date (Referee #1, Referee #3, and Referee #2). Referee comments are marked with RC and are in black. Author Responses are clearly labeled and are in blue.

Please see the attached Supplement to access the document with the detailed responses to the three referees compiled in one document.

Please also note the supplement to this comment:
https://www.atmos-chem-phys-discuss.net/acp-2019-1124/acp-2019-1124-AC4-supplement.pdf

**Supplement:**

**Response to Interactive comments on: "Long-term Brown Carbon and Smoke Tracer Observations in Bogotá, Colombia: Association to Medium-Range Transport of Biomass Burning Plumes" by -** Juan Manuel Rincón-Riveros et al.

We thank the three anonymous reviewers for their thoughtful feedback, and constructive comments, which undoubtedly helped to improve our manuscript. The three reviewers accurately pointed out that the manuscript was missing an uncertainty analysis associated with the reported eBC and BrC concentrations, and they all emphasized the need to perform a sensitivity analysis for the parameters involved in the attribution of BrC and Black Carbon, to make the calculations more transparent. We addressed these issues (and all the other comments).

This particular issue was addressed in the following way:

- Detailed description showing step-by-step the decomposition of absorption measurements ($b_{abs}$) due to fossil fuel (FF) and biomass burning (BB).
- Detailed discussion of uncertainties associated with our approach, both, from the decomposition into FF and BB (assumed angstrom exponents), and from uncertain values of mass absorption cross sections
- Sensitivity analysis to parameter choices.

All the additional comments were also addressed and incorporated in a new version of the manuscript. Detailed responses to each one of the reviewers' comments are detailed below (in blue) sorted by publishing date (Referee #1, Referee #3, and Referee #2). Referee comments are marked with **RC** and are in black. Author Responses are clearly labeled and are in blue.

**Anonymous Referee #1**

**RC:** Light-absorbing aerosols can affect both air quality and climate, so understanding their source and transport is important. This manuscript used a bunch of different observations to study the sources of light-absorbing aerosols over densely populated areas in the Central Andes of Northern South America. **It showed that these aerosols are closely related to medium-range transport of biomass burning plumes**. My comments are listed below.

Major comments

**RC:** I am concerned about the uncertainty associated with the BrC and BC measurements reported in this work. As mentioned in the work and reported by many other studies, there is large variability in reported mass absorption cross-section and Angstrom exponent values for absorbing aerosols. However, this study still used a single certain value for these variables (i.e. =7.77 g/m3; FF=1; BB = 2), without estimating the uncertainty due to the variation of these values. I expect that both eBC and BrC concentrations would change a lot if one assumes different values for these optical parameters. In addition, the authors should also estimate the uncertainties resulting from the process of measuring and analyzing the biomass burning tracers.

> **Authors Response:** We have now addressed the issue of uncertainty by performing sensitivity analysis on the parameters used in the calculations. Regarding the use of a (mass absorption cross section) MAC 7.77 g/m2 for eBC (at 880 nm), we would like to clarify that, by definition, it

is necessary to assume a specific MAC to convert $b_{abs}$ into a "Equivalent Black Carbon" concentration. We strictly followed the recommendations of Petzold et al., 2013 (Atmos. Chem. Phys., 13, 8365–8379, 2013) by explicitly stating the MAC used, so the calculation is transparent and reproducible. We now explicitly mentioned this in the manuscript. Furthermore, we have now included $b_{abs}$ in Figure 2 by adding a secondary axis.

One significant issue was the lack of sensitivity to parameters. **Figure R1** shows a sensitivity analysis performed on the parameters $\alpha_{FF}$ and $\alpha_{BB}$. We computed the inferred BrC concentration for each set of parameters for high (DJF) and low (JJA) BB activity periods. Because our data is strongly influenced by urban emissions (dominated by traffic in Bogota) our observed Angstrom exponent is on average close to 1. Therefore, our deconvolution is much more sensitive to the assumed value of $\alpha_{FF}$ than it is to the much more uncertain $\alpha_{BB}$. However, it should be noted that in all the parameter combinations the same trend remains, namely that during the high BB periods BrC is significantly higher than during JJA (i.e., has a strong seasonality). A discussion in this regard is now included in the manuscript and the sensitivity analysis included in the Supplementary Material.

[Figure]

**Figure R1** – Response to Reviewers – Parametric sensitivity

Minor comments

**RC:** Line 168: "The quartz filters were pre-baked at 550°C for 12 hours to reduce their organic background and later placed in." why is it needed to be heated? Wouldn't it reduce the biomass burning semi-volatile OA?

**Authors Response:** The filters are pre-baked before being deployed for sampling. This is done exactly as the reviewer points out, to reduce semi-volatile OA from the filters, reducing this way any potential artifact during analysis post-sampling. We clarified this in the manuscript, and it reads "Previous to sampling, the quartz filters were pre-baked…."

**RC:** Line 174. What is LOD?

**Authors Response:** We intended LOD to stand for "Limit of Detection". We now explicitly define the term in the manuscript.

**RC:** Line 173-179. It seems OC and EC are measured in the same way? Then how does one differentiate OC from EC?

> **Authors Response:** OC and EC are measured in the same instrument, with a technique called TOT (thermal-optical transmittance). However, they are not measured in the same way. The TOT measurement is based on the fact that the organic carbon contained in particles volatilizes at different temperatures. The organic carbon is defined in this technique as the carbon that becomes gas in a Helium atmosphere at temperatures below 580°C. Meanwhile, EC in this technique is defined as the fraction of carbon that does not volatilize after exposing it to 580°C, but that oxidizes when oxygen is added to the controlled atmosphere at temperatures above 580°C. The quantification of carbon in each case (either volatilized or oxidized) is done by converting it to CH4 to be detected with an FID.
>
> We now expanded the explanation to avoid any potential confusion.

**RC:** Line 236. "The similarity between both datasets shows that eBC measurements at the site are overwhelmingly dominated by EC emissions from urban traffic and industrial emissions". No absorbing OC emissions from urban traffic and industrial emissions?

> **Authors Response:** The phrasing was modified in this section. The phrase now reads **"*The strong correlation between both datasets suggests that eBC at the Monserrate site is closely associated to urban emissions. According to a recent emission inventory in Bogotá, mobile and industrial emissions are the dominant primary particle sources in the city. Furthermore, cargo and public transportation have the largest emissions share, and most of those vehicles are diesel powered (Pachón et al., 2018).*"**
>
> Regarding the question of -No absorbing OC from urban traffic and industrial emissions? - It is possible (as has been recently show in the literature) that fossil fuels contribute to UV absorbing carbon (i.e., BrC). We acknowledge this in the paper now (in the introduction). However, EC is known to be the main absorber at near IR wavelengths, while OC from fossil fuel combustion is not a particularly strong absorber of near-IR light.

**RC:** Line 250. I think the major reason for the seasonal pattern in PM2.5 is the different emission source/strength in different seasons.

> **Authors Response:** Indeed, as the reviewer points out, this is exactly our working hypothesis in this paper, namely that biomass burning emissions in the region increase $PM_{2.5}$ concentration during the months of January-to-April, and we believe that we demonstrated that through measurements of biomass burning tracers in different seasons. In that specific paragraph we were merely pointing out that there are also meteorological conditions during those months (stronger surface inversions, stable conditions, lower mixing heights) that could concurrently have an impact of increasing $PM_{2.5}$ concentrations (this is explained in the reference Mendez-Espinosa et al., 2019). Furthermore, there is no clear annual pattern in either public transport, cargo transport, or industrial activities. There is no seasonal change in fuel composition as does occur in other countries.

**RC:** Line 263. I don't understand the reasoning here.

**Authors Response:** Point well taken. What we intended to say here was that the BrC we detected was likely aged biomass burning (because the sources are located hundreds of km away from our measurement site). The intended message is conveyed in the next section. Therefore, those lines were removed from the manuscript.

**RC:** Line 302. Not clear to me how the authors get these numbers.

**Authors Response:** These numbers were obtained by averaging WSOC for high and low BB activity seasons respectively (i.e, those represented by the open and filled circles in Figure 4b). This now reads: *"The mean WSOC observed for low BB activity was 2.5µgCm⁻³ while for high-BB activity period was 4.2µgCm⁻³ reaching up to 8µgCm⁻³."*

**Anonymous Referee #3**

**RC:** This manuscript presents 3-year measurements of aerosol light absorption at multiple wavelengths over a site in the Northern South America (NSA) region. These measurements are combined with campaign-based biomass burning tracer measurements, MODIS fire counts and back-trajectory analysis to examine seasonal variations and source attributions of black carbon and brown carbon. **It is one of the few observational studies over NSA, and clearly demonstrates the influences of nearby biomass burning on the local air quality in densely populated areas. The long-term observations of biomass burning aerosol properties are also useful in revealing the regional and temporal variability in light absorbing aerosols. The sample collection and data postprocessing parts are well described**.

My major concern is about the inference of brown carbon concentration in section 2.2.
First, the assumptions of FF AAE (=1) and BB AAE (=2) are subject to large uncertainty. How sensitive are the derived BC and BrC concentrations to these assumed AAEs? It would be helpful to include some sensitivity analysis by varying the AAE values.

**Authors Response:** We have now included a sensitivity analysis showing how the uncertainties associated to these parameters impact the calculated attribution of absorption to combustion of biomass or fossil fuels (**Figure R1**). We also enhanced the discussion on the sources of uncertainty. In a now expanded supplementary material, we show the impact of parameter choice on the inferred BrC concentration. After performing this analysis, we showed the estimated BrC is only slightly affected by the choice of $\alpha_{BB}$, while it is more sensitive to $\alpha_{FF}$. However, in any case, the correlation of BrC and MODIS fire counts remains unchanged. Figure R1 and a subsequent discussion on the sensitivity is now included in the Supplementary Material.

For our specific data set, heavily influenced by traffic emissions , the observed angstrom exponent is closer to 1 ($\alpha_{450nm-950nm} = 1.025 \pm 0.2$ and $\alpha_{450nm-880nm} = 1.065 \pm 0.22$). Therefore, the inferred BB fraction is much more sensitive to $\alpha_{FF}$ than it is to $\alpha_{BB}$. This can be seen in the **Figure R1** of this response (which has also been included in the Supplementary material for the final manuscript). This is positive for our study, as it is well known that $\alpha_{BB}$ is

much more uncertain than $\alpha_{FF}$ (which is largely accepted to be $\simeq 1$). This analysis is now included in the manuscript.

**RC:** Furthermore, lines 156-157 indicate that BrC concentration is computed as the product of eBC (equivalent BC concentration) and f_BB (fractional contribution of biomass burning to absorption). This is confusing: isn't the product equal to BC concentrations from the BB sources? How is it related to the BrC concentration? Presumably, BB aerosols should include both BC and BrC. But the inference method of BrC in section 2.2 seems to imply that absorption in BB aerosols is due to BrC. The calculation of BrC concentrations needs clarification.

**Authors Response:** A section was included in the supplementary material to expand and clarify the decomposition method applied in our study. The Methods section was also expanded to improve clarity. The method we used (Sandradewi et. al. 2008) is often referred to as the "Aethalometer model". In our manuscript (section 2.2), absorption at any given wavelength is indeed considered to be due both to BB and FF at any given wavelength. The FF contribution is associated with a $\lambda^{-1}$ component (typical of BC rich FF sources) and the BB component is associated with an Angstrom exponent >1 (typical of sources with light absorbing OC, such as BB). Our approach, as suggested by the manufacturer, is to use optical properties of black carbon to estimate mass. This is likely an underestimation of true BrC mass as most studies suggest its mass absorption cross sections is lower than that of BC.

**RC:** Another suggestion is since there are previous studies of BrC from the Amazon BB region, it'd be interesting to compare the derived BrC loadings and absorption properties over NSA with those in discussions. That would help extend the findings in this study to a larger regional context.

**Authors Response:** Done. We included some new references were absorption measurements of BB aerosols are made in NSA. These include (Saturno et al., ACP, 2018; Hamburguer et al, ACP, 2013). The addition now reads *"Our observations are broadly consistent with other available studies of aerosol absorption in the region that have reported an increase in $b_{abs}$ and Angstrom exponent during the dry season. Observations at the ATTO tower in central Amazonia show $b_{abs,635nm}$= 4.0±2.2Mm⁻¹ during the dry season (Saturno et al., 2018). Other observations at Pico Espejo, in NSA show $b_{abs,525nm}$= 0.91±1.2Mm⁻¹ during dry season, corresponding to three times the mean value observed during the wet season. However, both sites correspond to locations near the source areas, while our observation site is an urban site far away from the main biomass burning areas"*.

Minor comments:

1. Line 38: the source of BrC is not limited to BB. They could also come from biofuel and biogenic sources. Suggest to revise the definition of BrC, i.e., Andreae and Gelencser, 2006

   **Authors Response:** Point well taken. We rephrased and reorganized this section to acknowledge other sources of BrC. It now reads: *"The organic material (OM) present in aerosol particles, mainly those produced in BB, biofuel combustion, and from other sources, has been recently shown to absorb light in UV and short visible wavelengths more efficiently than BC. The absorption increases proportionally to the amount of OM present in the aerosol (Yan et al.,*

*2017; Mkoma et al.,2013). The collection of UV light-absorbing organic compounds present in aerosol particles is often termed Brown Carbon (BrC) (e.g., Kirchstetter et al., 2004; Andreae and Gelencsér, 2006; Wang et al., 2018), which is also a contributor to radiative forcing.”*

2. Lines 39-40: This sentence is inaccurate. The referred paper Bond et al., 2013 suggests that BC is the second largest contributors to anthropogenic radiative forcing, not BB particles

**Authors Response:** This oversight is now corrected in the manuscript. We now use the reference more accurately. It now reads: *“Due to its optical properties, EC is sometimes measured through light-absorption techniques, and when measured this way is referred to as equivalent Black Carbon (eBC) (Petzold et al., 2013). BC is the second largest contributor to anthropogenic radiative forcing whit open burning of forests and savannas being the largest source (Stohl et al., 2015; Bond et al., 2013).”*

3. Line 65: missing a comma after "...their work”

**Authors Response:** Corrected.

4. line 66: replace "finding" with "indicating”

**Authors Response:** Corrected.

5. Line 83: "Levoglucosan" doesn't need an initial capital letter

**Authors Response:** Corrected.

Line 92: brown carbon and black carbon do not need initial letter capitalized. This needs to be corrected in other places as well

**Authors Response:** This is now corrected throughout the manuscript.

**RC:** Section 2.4: why not make the observatory site directly as the starting point of the back-trajectories, instead of Bogota? Since they are located at different altitudes.

**Authors Response:** The (lat, lon) coordinates used in the calculation are indeed those of the Monserrate site. This typo is now corrected. However, it should be noted that the spatial resolution of the meteorological data (1 degree, roughly equivalent to 110 km) is too coarse to accurately represent differences in back-trajectories starting from nearby points.

In a previous study, we found that due to the complex topography of the region, selecting starting points that are too close to or at the surface yields unrealistic back-trajectories. That is why we selected our arriving point at 1000 m.a.g.l, so it is not at the surface but remains within the mixing layer. This is now explicitly stated.

7. Line 123: W doesn't need capitalization

**Authors Response:** Corrected.

8. Line 126: what is Davis Advantage Pro II?

**Authors Response:** This is now corrected. The Vantage-Pro2 (it was erroneously typed in the original manuscript) is the specific model of the meteorological station used for the data collection, which is made by Davis Instruments. This is now explicitly written in the manuscript.

9. Figure 1 (b): suggest to add a color scale for the background map. Is it for terrain height?

**Authors Response:** Point well taken. The figure was modified and included a more descriptive legend (See **Figure R2** included in this response). The color scheme is related to land-use cover (urban area, hills, and cropland/grassland). The shading is intended to qualitatively show terrain height variations, and this is mentioned in the caption. If height contour levels are included the plot gets cluttered and then is no longer effective.

[Figure]

**Figure R2** – Response to Reviewers – Modified figure

**RC:** Line 209: what is the spatial resolution of GDAS1 meteorology?

**Authors Response:** This is now corrected. GDAS1 meteorology is 1° x 1°. We now explicitly mention this in the manuscript.

**Anonymous Referee #2**

**RC:** This paper investigates the contribution of biomass burning from distant locations to air quality in Bogota Colombia based on an extensive data set of aerosol light absorption at multiple wavelengths. Most data reported are from a measurement site upwind and at higher elevation than the city. Filter measurements of smoke tracers are also used to support the analysis, along with satellite-based fire counts and air mass back trajectories. **Overall the paper is a nice contribution to an understudied location and appropriate for publication in this journal. The results are interesting and the analysis very thorough**, however, some components are confusing and should be clarified.

I agree with the other two reviewers that the sensitivity of the reported results to the choice of AAE for BC (AAE=1) and for BrC (AAE=2) should be assessed. A value of BrC AAE of 2 seems especially

arbitrary. It is not clear to me why the authors utilized this analysis method at all since it adds unnecessary complexity and ambiguity; more related to this question follows below.

> **Authors Response:** This comment is common to all reviewers. We addressed the issue of uncertainty in two ways: 1- By expanding and explaining the uncertainty associated to the attribution of absorption to BB and FF (including sensitivity analysis) and 2- by discussing potential uncertainties associated to transforming $b_{abs}$ into eBC and BrC. Figures were modified to include an axis with $b_{abs}$ (in Mm$^{-1}$)
>
> We also expanded the supplementary material to clarify and make more transparent how the separation between BB and FF was performed.

**RC:** Why was 470 and 880 nm light absorption data used in the fractional biomass burning calculation? Explicitly state the reason. e.g, why not 370 and 950 nm, respectively? Similarly, why was 880 nm used for eBC, not 950 nm? Why not use all the wavelength data in some way, instead of just selecting a few wavelengths from the measurements (more on this below)?

> **Authors Response:** We did not use data from the 370 nm channel since its noise to signal ratio is higher than that of other channels. To analyze BrC/BC typically a short wavelength and a near-IR wavelength are used. The choice of 880 nm for BC was done as this is what has been historically reported in previous aethalometer studies. BC at 880 and 950 nm channels is almost identical for our data.
>
> This is now explicitly mentioned in the manuscript: "***The Angstrom exponent was computed using a wavelength in the near-UV, where absorption from some organic compounds can be significant, and a near IR wavelength, where absorption is dominated by black carbon. However, as the 370 nm channel had a larger noise to signal ratio, the limit of detection of this channel was considerably higher and was not used in the analysis. Equation 1 was then applied to $b_{abs}$ measured at 470 nm and 880 nm wavelengths to compute an observed α.***"
>
> We also performed some sensitivity analysis on the pair of channels used to calculate the Angstrom exponent and evaluated its impact on our results. The mean Angstrom exponent varies slightly according to the wavelength pair chosen ($\alpha_{450nm-950nm} = 1.025 \pm 0.2$ and $\alpha_{450nm-880nm} = 1.065 \pm 0.22$) affecting the absolute value of inferred f_BB and BrC. However, because most of our analyses are based on correlations and associations to fires, these remain unchanged.

**RC:** Why does one even need to calculate a BC and BrC concentration, instead of just using the absorption coefficient? For example, simply using the absorption measured at a high wavelength as a tracer for BC and Abs measured at a low wavelength (e.g., 370, or if too noisy, 470 nm) as a tracer for BrC, after the Abs by BC at that wavelength is removed. This can be done by assuming a BC AAE of some value, such as 1. This seems like a much more transparent way to apportion BC and BrC from the multiwavelength Aeth data and it eliminates the need to assume a characteristic BrC AAE. It also simplifies an uncertainty analysis on the sensitivity of the results to only the value of BC AAE. It would be interesting to see a correlation between the BrC mass inferred by the method in this paper and the BrC abs at some wavelength (eg, 370 nm).

> **Authors Response:** We are aware of the challenges of performing this decomposition. The method proposed by the reviewer is plausible, but it disregards contributions to absorption from BrC even at high wavelengths. The method we employed accounts for the contribution of BrC

and BC absorption at all wavelengths. We now discuss this much more thoroughly in the manuscript.

To address the valid concern regarding calculations of BC and BrC, we now added $b_{abs}$ ($Mm^{-1}$) in a secondary axis in Figure 2b and 2c. Since eBC and BrC are both proportional to $b_{abs}$, their correlation to the other datasets remains unchanged.

**RC:** Instead of picking a specific wavelength for BrC why not use all the Abs vs wavelength data. That is, fit the data with an AAE using all the wavelength and then use the fit to predict BrC AAE (data AAE-1) and then determine light absorption at some low wavelength with fit AAE-1.

**Authors Response:** During the data analysis phase, we did explore a variety of multi-wavelength methods. However, most of those methods require and even greater number of assumed parameters (e.g., Massabó et al., 2015). In their approach 5 wavelengths are used, but there is the need to assume three parameters and solve for three more. We tested this method and when applied to our data was much more sensitive to parameters choice than the simpler method we employed.

**RC:** Light absorption data are based on PM1, chemical composition and mass on PM2.5. PM1 was chosen to reduce possible influence of dust light absorption on the inferred BrC mass. The authors could test if there is any correlation between dust (eg, Ca2+) and BrC.

**Authors Response:** This is an excellent suggestion. However, we do not have Ca2+ data available in our samples. We think there is strong evidence showing that our BrC observations are strongly linearly correlated with levoglucosan and other BB tracers, suggesting its origin is indeed from biomass burning.

**RC:** Line 236-237: This line is unclear, suddenly there is a discussion that changes from eBC to EC. How does this data prove eBC is EC. Why not just say that eBC is from urban traffic and industrial emissions? Also, why is EC only assumed to be from these two sources?

**Authors Response:** Corrected. The phrase now reads "***The strong correlation between both datasets suggests that eBC at the Monserrate site is closely associated to urban emissions. According to a recent emission inventory in Bogotá, mobile and industrial emissions are the dominant primary particle sources in the city. Furthermore, cargo and public transportation have the largest emissions share, and most of those vehicles are diesel powered (Pachón et al., 2018).***"

**RC:** Line 248-249, first line after heading 3.1. This line is unclear. Is the eBC, BrC and fire counts data (Fig 3b) from the hill top site and the PM2.5 mass (Fig 3a) from the urban air quality stations in the city? That means that Fig 3a has data from two different sites? This complicates the comparison and the discussion that follows this line. More clarity is needed here. Please specify on the plots in Fig 3 what site the data is from.

**Authors Response:** This is correct. We tried to be as clear as possible in the caption and throughout the text. Now Figure 2 in the manuscript has been modified to include the origin of the data (i.e., panels (b) and (c) are marked Monserrate site). Similar changes were performed on Figure 3.

However, we want to emphasize to the reviewer that the aim of Figures 2 and 3 in the paper is to show that our BC measurements are strongly correlated to PM2.5 in the city, while BrC

measurements do not. Instead, BrC resemble regional fire counts according to their correlation coefficients.

**RC:** Line 282, typo change that to local emissions, to, than to local emissions.

**Authors Response:** Corrected.

**RC:** Line 289-290 states, … However, optical methods are not always quantitative methods to determine BB aerosol loading. What is this statement based on?

**Authors Response:** The statement originally meant to refer to the issue of translating absorption coefficient data into concentrations (which, as the reviewer pointed out, require the assumption of a MAC). That is what we originally meant by "quantitative". This whole paragraph is now rewritten and now reads: *"However, due to the uncertainties in mass absorption cross sections, aerosol absorption measurements are not always straightforward to translate into BB aerosol concentrations. To establish the relationship between our Aethalometer based BrC (Section 2.2) and analytical methods to quantify BB aerosols (Section 2.3), we compared…."*

**RC:** Line 314. Is this true; the Monserrate site (also called at times, the hill top site) maybe a fairly close distance to the urban center, but it is decoupled from the city at times due to its higher elevation and changes in BL height. This mixing of the hill top site with the urban site throughout the paper leads to confusion. Often the term monitoring site is also use, which is apparently the Monserrate site, not the urban air quality sites? I suggest being more specific and consistent throughout the paper on what the sites are called.

**Authors Response:** Point is well taken. Our monitoring site is now referred to as Monserrate Site consistently throughout the whole manuscript. We also included a paragraph in the Methods to make explicit that our data comes from two different sources: our station at Monserrate (for eBC, BrC, and smoke tracers), and the AQ monitoring sites in the city (for $PM_{2.5}$ only).

**RC:** Last line of Conclusions. What is the 13% based on, mass ratio of eBC and BrC. This is then not an optical ratio and should be noted, it may also depend on how BrC was determined (AAE=2). Again, calculating mass concentrations of BC and BrC from the absorption data just leads to confusion and more uncertainty, in my view.

**Authors Response:** The 13% is based on $f_{BB}$. With the expanded and improved Methods and Supplementary materials we show that $f_{BB} = b_{abs,BB}(\lambda)/b_{abs}(\lambda)$. We performed a sensitivity analysis of this monthly mean $f_{BB}$ with $\alpha_{BB} = 2.0 \pm 0.4$ and $\alpha_{FF} = 1.0 \pm 0.1$. We know report a range in this percentage. It now reads: *"During our observation period, the month with the largest contribution of BB aerosols to light-absorbing material was March with10%±5%. The largest contribution was identified for February and March 2019, with 13%±6%. The uncertainty estimates in this fraction are due to uncertainty on the assumed absorption Angstrom exponent for biomass burning and fossil fuel burning used in the attribution algorithm"*

---

## Author Response (AR2)

**Response to editor comments on:**

**"Long-term Brown Carbon and Smoke Tracer Observations in Bogotá, Colombia: Association to Medium-Range Transport of Biomass Burning Plumes" by -** Rincón-Riveros et al.

We acknowledge the editor comments, which touch on relevant aspects of the evolution of BB aerosols in the atmosphere. A revised version of the manuscript with highlighted changes is included. Here we enclose a brief reply to your comments:

**Editor Comments:**

**1.** Please check spellings and typos throughout the final Manuscript. For example, Caption to Figure 1 has word "terrain" repeated and also has a typo: "shoing" instead of "showing".

> **Response: We carefully proofread the manuscript to correct these typos.**

**2.** Please comment on influence of potential photo-bleaching of brown carbon in measurements as reported in several studies. Could bleaching timescales of brown carbon be quantified from the long-term measurements?

> **Response: We have now included a paragraph in the document discussing the potential implications of photo-bleaching for our measurements and included a recent reference on this topic (please see lines 333 to 340 in the manuscript with highlighted changes). We do not believe however that our measurements alone are suitable to quantify bleaching timescales. For that, it would likely be necessary to carry out chemical transport modelling to better represent atmospheric transport times for different air masses.**

> **The added paragraph reads (line 329 onwards):** *"However, recent observational studies of ambient BB particles have shown poor correlation between levoglucosan and brown carbon in aged BB plumes, likely due to BrC photobleaching and to the oxidation of levoglucosan (e.g, Wong et al., 2019b). The strong correlation in our samples might indicate atmospheric transport times of up to 2 days, consistent with the lifetime of levoglucosan (Henniganet al., 2010) and that of BrC, recently estimated between 13 to 30 hours (Wong et al., 2019b). These atmospheric transport times are consistent with the strong association between BrC and MODIS active fires within 600 km of the sampling site (Table 2)"*

**3.** Figure 4 shows that levoglucosan strongly correlates with brown carbon during high BB periods. Levoglucosan is also reported to degrade in the atmosphere with aging. Could the authors comment on how degradation of levoglucosan affects its correlations with brown carbon?

> **Response: We addressed this issue together with the photo-bleaching comment (please see lines 333 to 340 in the manuscript with highlighted changes) as discussed in the added paragraph.**

[revised manuscript text omitted]